# TUMIX: MULTI-AGENT TEST-TIME SCALING WITH TOOL-USE MIXTURE

**Yongchao Chen**[1,2,*]  **Jiefeng Chen**[3]  **Rui Meng**[3]  **Ji Yin**[1]  **Na Li**[2]
**Chuchu Fan**[1]  **Chi Wang**[4]  **Tomas Pfister**[3]  **Jinsung Yoon**[3]

[1]MIT    [2]Harvard    [3]Google Cloud AI Research    [4]Google DeepMind

yongchaochen@fas.harvard.edu    jinsungyoon@google.com

## ABSTRACT

While integrating tools like Code Interpreter and Search has significantly enhanced Large Language Model (LLM) reasoning in models like ChatGPT Agent and Gemini-Pro, practical guidance on optimal tool use is lacking. The core challenge is effectively combining textual reasoning, coding, and search for diverse questions. In this paper, we propose Tool-Use Mixture (TUMIX), an ensemble framework that runs multiple agents in parallel, each employing distinct tool-use strategies and answer paths. Agents in TUMIX iteratively share and refine responses based on the question and previous answers. In experiments, TUMIX achieves significant gains over state-of-the-art tool-augmented and test-time scaling methods, delivering an average accuracy improvement of up to 3.55% over the best baseline on Gemini-2.5-Pro and Gemini-2.5-Flash across key reasoning benchmarks, with near-equal inference costs. We find that agent diversity and quality are crucial and can be enhanced by using LLMs to auto-optimize agent designs. Furthermore, TUMIX can halt refinement upon reaching sufficient confidence, preserving performance at only 49% of the inference cost. Further scaling can achieve higher performance, albeit at a greater cost.

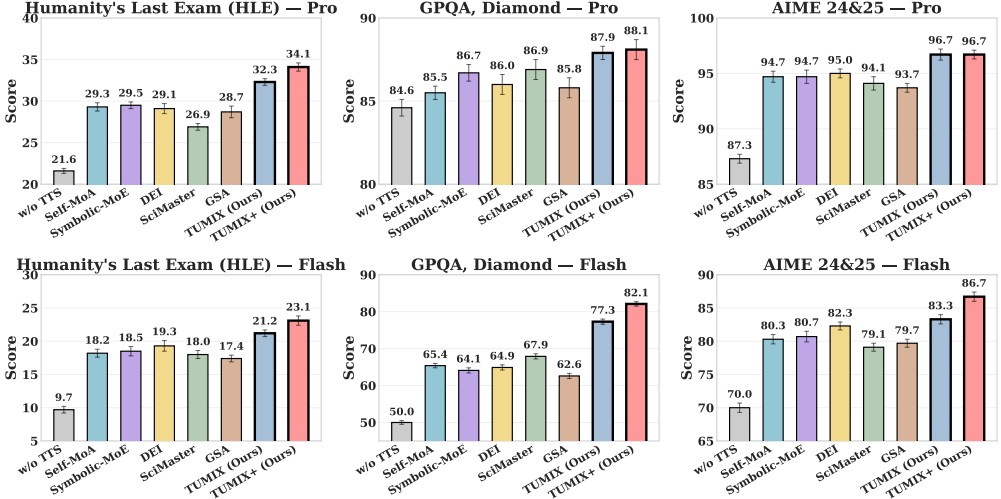

Figure 1: Comparison of tool-augmented test-time scaling methods on Gemini-2.5-Pro (first row) and Gemini-2.5-Flash (second row) across HLE, GPQA, and AIME 24&25. Except for methods without test-time scaling (w/o TTS) or additional scaling (TUMIX+), all methods in the same subplot use nearly the same number of inferences and tokens. For fair comparison, methods that originally lacked tool use are run with strong tool-augmented agents instead of text-only agents. Each score is the average of three repetitive runs.

---

*This work was done while Yongchao was a research intern at Google Cloud AI Research.

# 1 INTRODUCTION

While reinforcement learning-based fine-tuning has greatly improved LLM reasoning (Guo et al., 2025), models still struggle with seemingly simple tasks (Chen et al., 2024b). Such tasks are often better handled with code (Madaan et al., 2022; Chen et al., 2022) or search (Jin et al., 2025; Li et al., 2025b). Textual reasoning is strong in semantics and commonsense, but weak in precise computation and in accessing or updating the latest knowledge.

A key challenge is fully utilizing the potential capabilities of textual reasoning, coding, and searching when facing distinctive questions with varied characteristics. Most input questions lack explicit cues for the best approach, and the combined text/code/search solution space is large. Frontier LLM-powered products such as ChatGPT, Claude, Gemini, and Grok report using code and search at test time to augment reasoning, but without publishing detailed methods. Recent work (Chen et al., 2024b) shows that current Code Interpreter implementations in OpenAI models often fail to balance text and code, leaving coding capabilities underused, as shown in Appendix Fig. 11. Moreover, public research still lacks a clear understanding of how to integrate Code Interpreter and Search for improved LLM reasoning.

To better leverage both tool use and LLM self-reasoning, we propose Tool-Use Mixture (TUMIX), a framework that integrates Code Interpreter and Search into LLMs via test-time scaling. TUMIX runs multiple diverse agents in parallel, each with different tool-use strategies. Their outputs are iteratively aggregated and refined across multiple rounds. In each round, every agent generates a new solution by considering both the original question and the previous round's reasoning and answers from all agents. TUMIX uses diverse agents and tool-augmented reasoning strategies to explore a wide range of possible solutions. The following iterative process encourages diverse reasoning paths and deeper integration. This design is inspired by prior test-time scaling methods such as Mixture-of-Agents (MoA) (Wang et al., 2024), which rely on multiple LLMs within a single framework and do not incorporate external tools. In contrast, TUMIX employs a single LLM with both text-only and tool-augmented agent frameworks, making it more generalizable for practical applications. Furthermore, in tool-augmented multi-agent test-time scaling, we find a diverse group of agents outperforms repeated use of the single best agent, a conclusion that differs from MoA (Li et al., 2025a). We later reveal that human pre-designed agent group can be further optimized by querying LLMs to self-design more diverse high-quality agents based on current ones, adding an average 1.2% improvement without cost increase.

Since questions vary in difficulty, they require different amounts of iterative refinement. We query the LLMs to decide whether to terminate refinement early, while still enforcing a minimum number of rounds to maintain answer quality. This adaptive early-termination strategy reduces inference costs to 49% of the original two settings (termination in a fixed round number or by majority-vote consistency across rounds), while preserving or even improving performance. The improvement arises because over-refinement rarely changes the final result and can even degrade performance, as correct answers may be mistakenly discarded.

Compared to the model without test-time scaling, TUMIX delivers an average +7.8% and +17.4% accuracy gains in benchmarks Humanity's Last Exam (HLE) (Phan et al., 2025), Graduate-Level Google-Proof Q&A (GPQA, Diamond) (Rein et al., 2024), and American Invitational Mathematics Examination (AIME 24&25) with base models Gemini-2.5-Pro and Gemini-2.5-Flash, respectively. Under the same inference costs, TUMIX also outperforms existing representative test-time scaling methods such as Self-MoA, Symbolic-MoE, DEI, SciMaster, and GSA, with an average +3.55% lifting compared to the best performing baselines. Notably, with further scaling, TUMIX raises Gemini-2.5-Pro accuracy on HLE from 21.6% to 34.1%, surpassing Gemini-2.5-Pro Deep Research at 26.9% (32.4% with higher compute) (Comanici et al., 2025). Test-time scaling hinges on two stages (Brown et al., 2024): (1) generating diverse candidate solutions and (2) selecting the correct one. For questions with both small answer spaces (e.g., multiple-choice) and large ones, diverse sampling greatly improves coverage. While it achieves high coverage on HLE (among generated answers in the whole round, at least one is correct on $\geq 65\%$ of questions), accuracy plateaus at about 34% because LLMs struggle to identify the correct answer among noisy candidates. We identify and explore four key factors: agent quality, agent diversity, refinement termination, and answer selection. Our work makes the following contributions:

1. **TUMIX: A competitive tool-augmented test-time scaling method.** We propose TUMIX, a novel framework for test-time scaling that integrates tool augmentation. Extensive experiments

demonstrate that `TUMIX` consistently outperforms strong baselines, achieving an average improvement of +3.55% over the best-performing prior methods.

2. **Key factors and mechanisms in tool-augmented scaling.** We provide a systematic analysis that distinguishes tool-augmented scaling from traditional test-time scaling:

- *Agent diversity and quality outweigh scale alone.* High-temperature sampling increases coverage, but heterogeneous agent strategies yield higher accuracy and lower cost than repeatedly sampling from a single best-performing agent.
- *Tool augmentation boosts performance.* Agent groups equipped with tools such as Code Interpreter and Search achieve superior coverage and accuracy compared to text-only agent groups.

3. **LLMs as agent designers.** We show that prompting LLMs to automatically generate diverse, high-quality agents based on existing ones further improves `TUMIX`. This yields an additional average accuracy lift of +1.2%.

4. **LLM-as-Judge for refinement termination.** We introduce an LLM-based judge to adaptively determine the optimal stopping round in iterative refinement. This prevents excessive refinement, which reduces diversity and can mistakenly discard correct answers. By enforcing a minimum refinement depth and querying the judge for termination, we achieve near-optimal accuracy while reducing inference cost to ~49% of the original.

## 2 RELATED WORK

**Code Interpreter and Search** Many benchmark tasks can in fact be better solved through code (Gao et al., 2023) and search (Li et al., 2025b), and recent work extends coding to reasoning and semantic analysis (Li et al., 2023a; Weir et al., 2024). Most prior approaches use either text (Yao et al., 2024) or code (Bairi et al., 2024; Zhou et al., 2023) exclusively as output. Recent work (Chen et al., 2024b) emphasizes the need to dynamically switch between modalities, proposing CodeSteer (Chen et al.) as a guidance model. Extensions with retrieval (Jin et al., 2025; Li et al., 2025b) and tool use (Qian et al., 2025) further improve reasoning, but lack the thorough exploitation of Code Interpreter and Search tools. Leading models such as OpenAI's ChatGPT Agent, Google's Gemini-Pro (Comanici et al., 2025), and XAI's Grok4 report using code and search at test time to augment reasoning, but without publishing detailed methods. Open work such as ToRL (Li et al., 2025c) and ReTool (Feng et al., 2025) investigates training reasoning models to integrate with Code Interpreters. However, their training and evaluation are limited to math problems, leaving a significant gap from real-world applications that demand effectiveness across broader benchmarks. ToolRL (Qian et al., 2025) instead focuses on teaching models to select among multiple tools, where the generated codes and search queries are relative simple and the evaluation tasks require less reasoning capabilities. SciMaster (Chai et al., 2025) samples the same pre-designed tool-use agent five times, then uses other pre-designed agents to critique, refine, and aggregate the answers. This approach shows clear improvement over single-inference text-only baselines, but the extent and manner of tool exploitation remain underexplored. In summary, integrating Code Interpreter and Search into LLM reasoning is essential and challenging. The academic community currently lacks methods and studies that fully exploit the benefits of LLM self-reasoning, code execution, and search, which is the focus of us.

**Test-time scaling** LLM self-exploration, reflection, and evaluation can enhance task performance across domains (Yang et al., 2022; Welleck et al., 2022; Madaan et al., 2023). Models like OpenAI o1 (Jaech et al., 2024) and DeepSeek R1 (Guo et al., 2025) showcase agentic behavior via Chain-of-Thought (CoT) reasoning and self-reflection, which is learned by RL-based training with rule-based outcome rewards (Shao et al., 2024; Wei et al., 2025). Apart from the training-based scaling, many research also explore scaling during LLM inference time by pre-designing prompt and agent frameworks. In these works, multi-agent reasoning has emerged as a promising paradigm for enhancing complex problem-solving and decision-making in AI systems (Wu et al., 2023; Li et al., 2023b; Topsakal & Akinci, 2023). Prior work finds gathering the answers from different LLMs improves LLM performance (Du et al.). Mixture-of-Agents (MoA) (Wang et al., 2024) further extends this idea by sharing and gathering answer among LLMs. However, Self-MoA (Li et al., 2025a) argues that LLM diversity may not be critical since replacing different types of LLMs with the best one achieves better performance. Symbolic-MoE (Chen et al., 2025b) further assigns different questions with different specialized LLMs. Instead of using different types of LLMs, many works such as DEI (Zhang et al., 2024), GSA (Li et al., 2025d), and SETS (Chen et al., 2025a) employ different agents from the same LLM for extensive test-time scaling, in which the agent types and

Table 1: 15 pre-designed agents used in `TUMIX`.

| Full Name | Short Name | Description (15 agents) |
| --- | --- | --- |
| `w/o TTS` | `Base` | Direct prompt. |
| `CoT Agent` | `CoT` | Chain-of-Thought prompt (Wei et al., 2022). |
| `CoT-Code Agent` | $\text{CoT}_{\text{code}}$ | CoT prompt to output code. |
| `Search Agent` | `S` | Uses WebSearch (LLM inherent tool only). |
| `Code Agent` | `C` | Uses Code Interpreter (base version). |
| `Code Agent+` | $\text{C}^{+}$ | Uses Code Interpreter (hinted version with extra human pre-designed priors). |
| `Dual-Tool Agent` | `CS` | Uses Code Interpreter + WebSearch (with 3 search variants). |
| `Guided Agent` | `CSG` | Dual-Tool agent (`CS`) *guided* by a steering module (Chen et al.) (with 3 search variants). |
| `Guided Agent+` | $\text{CSG}^{+}$ | *Guided* agent (`CSG`) with enhanced/hinted prompts (with 3 search variants). |

frameworks are explored (Chen et al., 2024a). Similar to our work, previous work in test-time scaling also finds the correct answer selection (Brown et al., 2024) is the main bottleneck. While previous work in test-time scaling do not incorporate tool-use of Code Interpreter and Search, we study how to utilize test-time scaling methods to better exploit the benefits of each reasoning mode.

## 3 TOOL-USE MIXTURE

Appendix B presents the full TUMIX algorithm, and Appendix C lists all agent prompts.

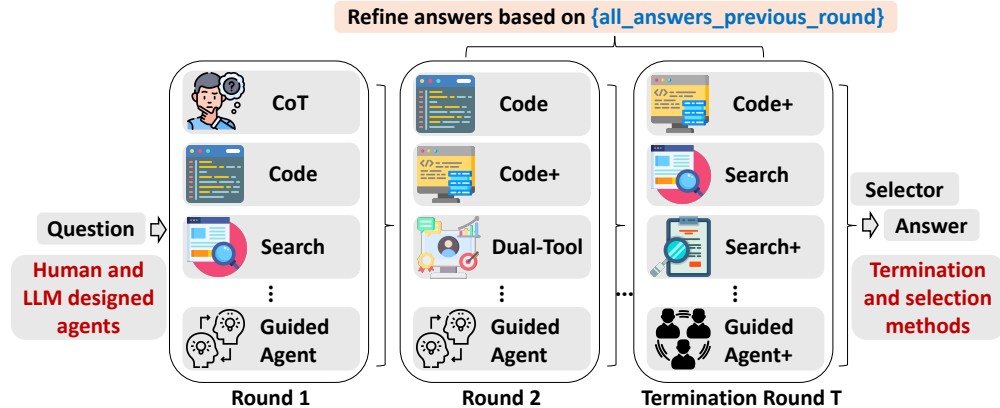

Figure 2: Overview of `TUMIX` framework. At each iteration, the responses from all agents in the previous round are concatenated with the original question, forming a joint prompt for the next round. This prompt is then provided to all agents (either the same or new agent groups) to produce refined answers. The subsequent prompts follow the structure illustrated in the above. The design of the agents, their number and specialization, the refinement termination criteria, and the selection strategies are the key factors that determine the effectiveness of the framework.

### 3.1 PRE-DESIGNED DIVERSE AGENTS

As shown in Fig. 2, we regard `TUMIX` as sequential decision-making under a compute budget with diverse and correlated experts (agents). Each round selects which agents to run, what they may read (communication policy), when to stop (optimal stopping), and how to aggregate (decision rule), trading off accuracy and cost. Let $q$ be a task with unknown correct answer $a^{\star}$ in answer space $\mathcal{A}$. There is a pool of agents $\mathcal{S} = \{s_1, \ldots, s_K\}$. Agent $s_i$ outputs an answer $Y_i \in \mathcal{A}$ at cost $c_i$ and has competence $p_i(q) = \mathbb{P}\{Y_i = a^{\star} \mid q\}$. Let $Z_i = \mathbf{1}\{Y_i = a^{\star}\}$ denote correctness indicators. Their dependencies (and hence ensemble diversity) are captured by a correlation or mutual-information structure over $\{Z_i\}$.

A policy $\pi$ (our focus) chooses in each round: (i) which agents to run, (ii) the communication graph (what each agent may read from prior rounds), (iii) the stopping rule, and (iv) the aggregation rule

producing $\hat{a}_\pi$. A canonical objective is

$$\max_\pi \; \mathbb{P}\{\hat{a}_\pi = a^\star\} \; - \; \lambda \cdot \text{Cost}_\pi, \tag{1}$$

where $\lambda > 0$ trades off compute and accuracy. In our work, the $\text{Cost}_\pi$ is the total number of inference times and input and output tokens to generate the final answer. In the default `TUMIX` setting, we utilize the same 15 pre-designed agents in all answer refinement rounds. These 15 agents have distinct reasoning and tool-use strategies, as summarized in Table 1. Agents with search access have three search methods (Google Search API (`gs`), inherent LLM search function (`llm`), or their combination (`com`)), yielding three variants per agent. For agents employing multi-round interactions with Search or Code Interpreter, the maximum tool interaction round number is set to 5. In Section 5.3, we discuss how to further query LLMs to automatically optimize and design more diverse agents to achieve better performance. We also compare with a dynamic setting where agent types vary across rounds.

## 3.2 REFINEMENT AS MESSAGE PASSING (ACCURACY RISES AND DIVERSITY SHRINKS)

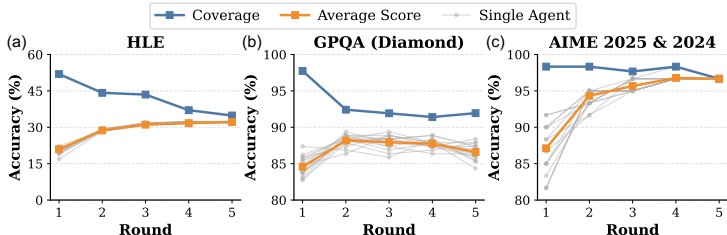

Figure 3: Evolution of coverage, individual agent scores, and average scores across refinement rounds. Coverage decreases monotonically across all benchmarks. For HLE and AIME, the average score rises over the initial rounds before plateauing. For GPQA, the average score improves early on but subsequently declines with further refinement.

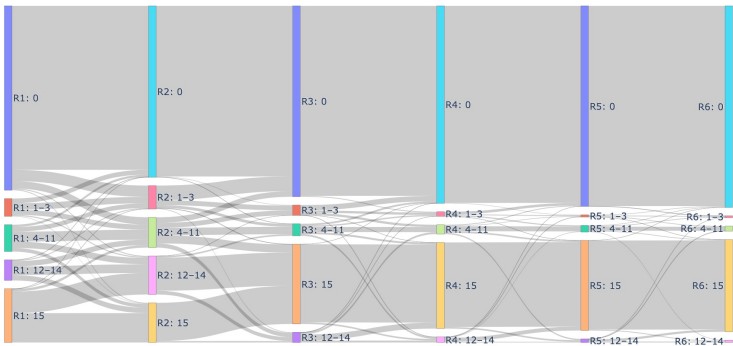

Figure 4: Sankey diagram of the evolution of correctly answering agents across 2,500 HLE questions over refinement rounds. Based on the distribution dynamics, we define five categories: all wrong (0), few correct (1–3), moderate correct (4–11), high correct (12–14), and all correct (15).

In each round, every agent independently generates a new solution by considering both the original question and the solutions provided by all agents in the previous round, as shown in Fig. 2. We evaluate the refinement process using two metrics: average accuracy and coverage (the probability of at least one correct) across agents in each round, which capture the quality and diversity of group answers (Brown et al., 2024). For a set $S \subseteq \mathcal{S}$, the coverage is

$$Coverage(S) \; = \; \mathbb{P}\Big(\bigcup_{i \in S} \{Y_i = a^\star\}\Big). \tag{2}$$

Under independence, $Coverage(S) = 1 - \prod_{i \in S}(1 - p_i)$. With positive correlations, $Coverage(S)$ shrinks. Fig. 3 shows the typical evolution dynamics of coverage, individual agent accuracy, and average scores over refinement rounds. Across all three benchmarks, coverage steadily declines, indicating that some correct answers are mistakenly discarded during iterative refinement. Note that from round 4 to round 5 in GPQA, coverage slightly lifts. When we extend the refinement to

rounds 6 and 7, the coverage decreases again as expected, confirming that the observed anomaly is transient. For HLE and AIME, the average score rises in the early rounds and then plateaus, while for GPQA it improves from round 1 to 2 but later declines. Fig. 4 visualizes the dynamics over 2,500 HLE questions. From round 1 to 2, the number of partially correct cases (few/moderate/high correct) increases, while both all wrong and all correct cases decrease. This suggests that initial thought-sharing broadens exploration and promotes diversity. After round 2, however, partially correct cases diminish toward near zero, while all wrong and all correct cases grow. This indicates that agents gradually converge to a single shared answer across rounds, either correct or incorrect.

### 3.3 TERMINATION IN OPTIMAL ROUNDS AND FINAL ANSWER SELECTION

The observed evolution indicates that round-by-round refinement not only improves answers in the initial rounds but also drives convergence, as each agent selects based on prior responses. However, due to the limited reasoning ability of LLMs, many correct answers are prematurely discarded. Beyond the early rounds, refinement rarely yields further accuracy gains and, in some cases, even degrades performance. Thus, identifying an effective termination strategy is essential for both robust performance and cost efficiency. Let $A_r$ denote acquired accuracy after round $r$. Define the expected marginal value of another round

$$\Delta_r = \mathbb{E}[A_{r+1} - A_r \mid \text{signals up to round } r]. \tag{3}$$

Stop at the first round $r$ where $\Delta_r \leq \lambda \cdot$ marginal cost (here is increased inference costs). A practical termination strategy decides whether to stop based on the estimated future gain $\Delta_r$, which relies on round-$r$ statistics such as (i) diversity collapse (coverage drop; rising agreement), (ii) vote margin between top answers, and (iii) answer entropy.

In `TUMIX`, our termination determination strategy is to query the LLM to decide whether to stop refinement and finalize answers based on the current round, with a minimum round number of 2. We find this termination strategy achieves nearly the same performance with only 49% of the inference cost. In Section 5.2, we explore other termination methods such as stopping once the majority answer stabilizes across two consecutive rounds or termination based on LLM confidence scores (Fu et al., 2025), but find only worse performance. After termination, we obtain the final answer through majority voting over the agents' responses, with Gemini-2.5-Pro selecting the most consistent output.

## 4 EXPERIMENTS

### 4.1 EXPERIMENTAL SETTINGS

**Benchmarks**  For a comprehensive evaluation and comparison across methods, we conduct experiments on three representative benchmarks that demand extensive reasoning and planning, particularly the ability to effectively leverage Code Interpreter and Search. **HLE** (Phan et al., 2025) consists of 2,500 highly challenging questions spanning diverse subject areas, including mathematics, biology, engineering, computer science, and the social sciences. It is designed as a final, closed-ended benchmark of broad academic capability. We evaluate both its text-only and multimodal subsets. In the following sections, we primarily use HLE to study different mechanisms, as it contains a large number of questions spanning diverse domains and is the most challenging benchmark. **GPQA** (Rein et al., 2024) is a multiple-choice dataset authored by domain experts in biology, physics, and chemistry. We focus on its most widely used subset, **GPQA Diamond**, which contains 198 of the most challenging and carefully curated questions. Finally, **AIME 2024&2025** comprises 60 problems from the 2024 and 2025 AIME exams, a notoriously difficult high school mathematics competition. All reported results are averaged over three independent runs.

**Baselines/`TUMIX` ablations and extensions/Test models**  As shown in Appendix Table 11, we compare against the following methods: (1) `Majority-Vote` (Brown et al., 2024);(2) `GSA` (Li et al., 2025d); (3) `Self-Reflection` (Ji et al., 2023); (4) `SETS` (Chen et al., 2025a); (5) `Self-MoA` (Li et al., 2025a); (6) `Symbolic-MoE` (Chen et al., 2025b); (7) `DEI` (Zhang et al., 2024); (8) `SciMaster` (Chai et al., 2025). For baselines (1)–(4), we use the `CS` agent, which has full access to both Code Interpreter and Search and achieves relatively high first-round accuracy (Appendix Table 19). For baselines (5)–(7), we select agents following their original methods. For `SciMaster`, we retain the original prompts and agents to ensure consistency with published results. We match the total inference counts of all baselines to `TUMIX` by adjusting agent numbers and

sampling repetitions for fair comparison. All the baselines have full access to Code Interpreter and Search. We evaluate `TUMIX` variants with different design choices, either to ablate framework components or to introduce improvements over existing `TUMIX`, as shown in Appendix Table 20. `TUMIX+` uses higher inference costs to test scaling effects, while other variants consume nearly the same inference and token counts. We evaluate our methods on the reasoning LLM Gemini-2.5-Pro and Gemini-2.5-Flash.

**Evaluation protocol** Answers are evaluated against ground-truth solutions, with Gemini-2.5-Pro assisting in normalizing answer formats when necessary or serving directly as the judge for answer comparison. In cases where the model outputs code as the final answer, we extract the code using predefined algorithms and execute it to produce the final result. To avoid infinite loops, all code execution whether during intermediate or final rounds is limited to 60 seconds. If execution exceeds this limit, a "code runtime error" is returned to the model for regeneration in intermediate rounds; in the final round, the task is marked as a failure. We report success rate as the primary evaluation metric. In addition to task performance, we also analyze token usage and inference time for each method in later sections.

## 4.2 OVERALL BETTER PERFORMANCE

Table 2: Experimental results of baseline and proposed methods on HLE, GPQA, and AIME 24&25. Except for the single-inference `w/o TTS` and the scaled-up `TUMIX+`, all methods use comparable inference costs for scaling. For some methods, Gemini-2.5-Pro's HLE results are used to select agents within their agentic framework. In these cases, the method has prior knowledge of HLE and the results cannot be strictly regarded as test performance. Such cases are marked with [*] in the HLE results. All the values are the average of three repetitive runs.

| METHODS | BASELINE METHODS | | | | | | | | | PROPOSED METHODS | | | |
|---|---|---|---|---|---|---|---|---|---|---|---|---|---|
| ACCURACY % | W/O TTS | MAJORITY VOTE | SELF-MOA | SYMBOLIC-MOE | DEI | SELF-REFLECTION | SETS | SCIMASTER | GSA | TUMIX | TUMIX-FIXEDR | TUMIX-EVOLVE | TUMIX+ |
| **GEMINI-2.5-PRO** | | | | | | | | | | | | | |
| HLE | 21.6 | 28.4 | 29.3[*] | 29.5[*] | 29.1[*] | 23.5 | 27.9 | 26.9 | 28.7 | 32.3 | 32.4 | 32.7[*] | 34.1 |
| GPQA | 84.6 | 84.9 | 85.5 | 86.7 | 86.0 | 84.9 | 85.3 | 86.9 | 85.8 | 87.9 | 86.8 | 88.1 | 88.3 |
| AIME 24&25 | 87.3 | 94.3 | 94.7 | 94.7 | 95.0 | 88.3 | 94.7 | 94.1 | 93.7 | 96.7 | 95.6 | 96.7 | 96.7 |
| **AVE. NORM.** | **64.5** | **69.2** | **69.8** | **70.3** | **70.0** | **65.6** | **69.3** | **69.3** | **69.4** | **72.3** | **71.6** | **72.5** | **73.0** |
| **GEMINI-2.5-FLASH** | | | | | | | | | | | | | |
| HLE | 9.7 | 17.9 | 18.2 | 18.5 | 19.3 | 10.4 | 18.5 | 18.0 | 17.4 | 21.2 | 20.9 | 21.9 | 23.1 |
| GPQA | 50.0 | 63.1 | 65.4 | 64.1 | 64.9 | 53.2 | 63.2 | 67.9 | 62.6 | 77.3 | 76.8 | 79.8 | 82.1 |
| AIME 24&25 | 70.0 | 80.0 | 80.3 | 80.7 | 82.3 | 72.3 | 74.0 | 79.1 | 79.7 | 83.3 | 83.3 | 86.7 | 86.7 |
| **AVE. NORM.** | **43.2** | **53.7** | **54.7** | **54.4** | **55.5** | **45.3** | **51.9** | **55.0** | **53.2** | **60.6** | **60.3** | **62.8** | **64.0** |

Table 2 shows that `TUMIX` outperforms all baselines, with average accuracy improvements of 2.0% and 5.9% over the best methods using Gemini-2.5-Pro and Gemini-2.5-Flash, respectively. Its superior performance over methods without answer sharing (`Self-Reflection`, `SETS`) highlights the importance of answer sharing in multi-round test-time scaling. Comparisons with methods lacking multi-round refinement (`Majority-Vote`, `Symbolic-MoE`, `DEI`, `GSA`) demonstrate the benefits of refinement, while comparisons with methods lacking agent diversity (`Self-MoA`, `SciMaster`) confirm the value of diverse agents. The accuracy improvement of `SciMaster` on HLE is smaller than reported by the authors. We suspect this discrepancy arises from differences in tools, as their Search and Code Interpreter modules are not open-sourced. In Appendix Table 12, we report both mean and standard deviation values. The performance gains of `TUMIX` over the strongest baselines exceed the reported deviations, indicating stable and consistent improvements. Additional experiments with diverse LLM types (Appendix Table 17) further confirm the robustness and generality of these results. To validate the statistical reliability of these improvements, we also perform two-tailed paired $t$-tests using repeated run scores, as shown in Appendix Table 13. Across nearly all benchmarks

and models, the resulting $p$-values are below 0.05, confirming that the improvements of `TUMIX` are statistically significant.

## 5 DISCUSSION

### 5.1 AGENT DIVERSITY AND QUALITY ARE CRITICAL

To investigate the role of agent diversity and quality in `TUMIX` performance, we compare groups of agents with varying levels of diversity and capability, as shown in Fig. 5 and Appendix Table 21. Under the same amount of refinement rounds and inferences, increasing the number of agents from 1 to 3 to 15 leads to substantial improvements in both coverage and average score across rounds on HLE and GPQA, indicating that diversity significantly benefits performance. Moreover, comparing a single strong agent with a single weak agent (see Appendix Table 19, where $CS_{gs}$ achieves higher first-round scores than `w/o TTS`), we observe that higher-quality agents consistently yield better coverage and higher average scores.

**Code Interpreter and Search increase answer diversity.** In Fig. 6, we evaluate three settings where each agent group consists of three agents, each sampling five times per round. The groups differ in their tool access: in Code_Text, agents cannot access Search; in Search_Text, they cannot access the Code Interpreter; and in Code_Search_Text, agents have full access to both. While the average agent quality (as measured by first-round scores in Appendix Table 19) is comparable across groups, the group with access to both Code Interpreter and Search achieves notably higher coverage and average scores. This result demonstrates that integrating complementary tools within agents enhances both reasoning and answer diversity, thereby facilitating more effective problem solving.

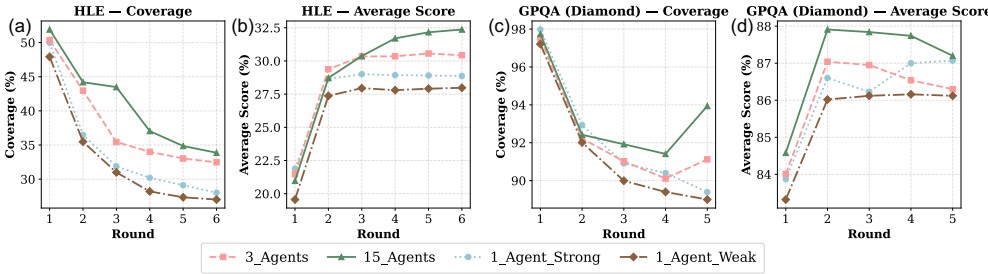

Figure 5: Coverage and average score vs. rounds under varying agent diversity and quality. In 15_Agents, all 15 varied pre-designed agents generate one answer each round. In 3_Agents, three strong agents ($C^+$, $CS_{gs}$, and $CSG_{gs}$, top-3 performed agents in round 1 as demonstrated in Appendix Table 19), each samples 5 times per round. In 1_Agent_Strong and 1_Agent_Weak, $CS_{gs}$ and `w/o TTS` sample 15 times, respectively.

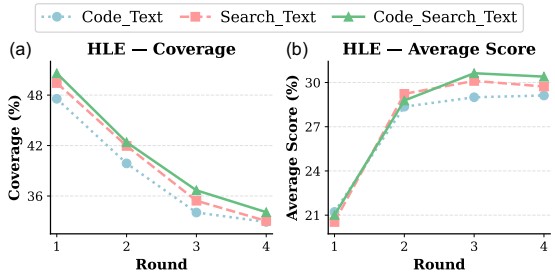

Figure 6: Comparison of groups all with three agents but either partial or full accesses to textual reasoning, coding, and search: Code_Text (`CoT`, `C`, $C^+$), Search_Text (`CoT`, `S`, $CS_{gs}$), and Code_Search_Text ($CS_{gs}$, $C^+$, and $CSG_{gs}$).

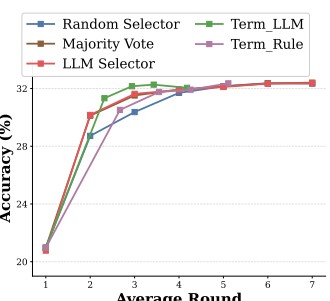

Figure 7: Comparison of refinement termination and answer selection strategies for higher accuracy with lower costs.

### 5.2 TERMINATION AND SELECTION METHODS

**Achieving optimal performance at 49% cost.** Tasks of varying difficulty require different numbers of refinement rounds, and excessive refinement can even degrade accuracy (see Fig. 3b). Thus, an effective termination strategy is essential to balance performance and cost. We evaluate two

termination strategies (Sec. 3.3): 1) **Term_LLM**, which queries the LLM to decide when to stop refinement, subject to a minimum round constraint; and 2) **Term_Rule**, which stops once the majority answer stabilizes across two consecutive rounds, also with a minimum round constraint. We vary the minimum number of rounds to examine how performance evolves as the number of rounds increases in Fig. 7, and we compare their peak performance in Appendix Table 21. Term_LLM achieves nearly the same peak accuracy as unlimited refinement, but with substantially fewer rounds. On average, Term_LLM retains optimal performance while requiring only 49% of the LLM inferences needed to obtain the final answer (LLM judging cost counted). The token costs are even less (approximately 46%), as the number of inference tokens used in later rounds exceeds that of the first two rounds. This demonstrates the effectiveness of using LLM-as-Judge to determine when refinement is sufficient and answers can be finalized. However, Term_LLM still requires a minimum number of refinement rounds (set to two across all benchmarks). This is because we observe that LLMs tend to be overconfident and may terminate refinement early, even when additional refinement could improve performance. Appendix Section F specifically discusses the impacts of minimum refinement round on TUMIX performance and efficiency.

For answer selection, we compare three strategies: (1) randomly choosing one agent's answer, (2) majority voting, and (3) LLM-based selection with LLM-as-Selector. Fig. 7 shows that majority voting and LLM-based selection consistently outperform random choice, especially in early rounds when agent answers diverge. However, once answers converge in later rounds, all selection methods yield similar results, and their impact becomes negligible. The multi-round refinement process is also a selection process. We also explore improved selection based on LLM token confidence (Fu et al., 2025), but observe no significant differences (Appendix Fig. 13).

### 5.3 HUMAN PRE-DESIGNED AGENTS VS. LLM GENERATED AGENTS

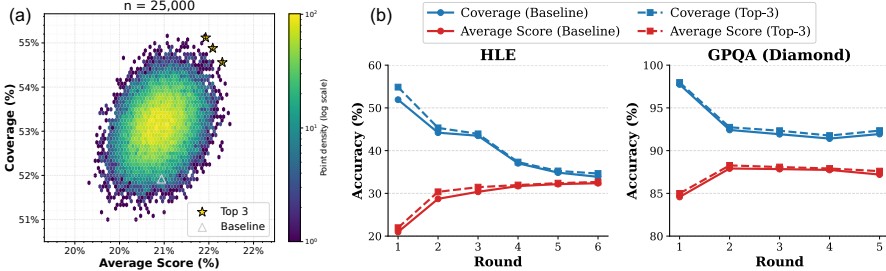

Figure 8: We evaluate the coverage and average score of 25,000 15-agent combinations sampled from 30 agents (15 pre-designed and 15 LLM-generated). Baseline refers to the original 15 pre-designed agents. Top-3 refers to the three sampled combinations with the highest joint performance in coverage and average score.

The human-designed agents and their tool-use strategies are built on existing frameworks or intuition. To explore whether stronger agents can be discovered automatically, we query Gemini-2.5-Pro with current agent code examples (total input prompt in Appendix Table 5) and ask it to generate full implementations of more diverse and high-quality ones, where the agent prompts and frameworks are all determined by LLMs. This yield 25 diverse agents beyond the 15 human-designed ones. From these, we retain the 15 that perform best in HLE with first-round answer generation. We then combine the 15 human-designed and 15 LLM-generated agents into a pool of 30, randomly sample groups of 15, and evaluate their average score and coverage (Fig. 8a). Compared to the baseline of 15 human-designed agents (gray triangle), many mixed groups achieve both higher average score and coverage. We select the top-3 groups based on the combined metric

$$\text{Combined Score}_i = \frac{\text{Coverage}_i}{\mathbb{E}[\text{Coverage}]} + \frac{\text{Average Score}_i}{\mathbb{E}[\text{Average Score}]}. \quad (4)$$

As shown in Fig. 8b, these groups outperform the original TUMIX in both HLE and GPQA. This demonstrates that increasing agent diversity and quality improves effectiveness, and that LLM-generated agents hold strong potential for further enhancing TUMIX. Appendix Table 22 describes each generated agent, whose strategies differ substantially from the original ones beyond prompt variations. Appendix Table 23 presents the agents in each top-3 group, with roughly half overlapping with the original group.

**Evolve agents in each round to enhance the diversity**   In all previous experiments, the agent set remained fixed across refinement rounds. We now investigate whether dynamically varying agent types per round can improve performance. As shown in Appendix Table 21, the variant `TUMIX-EvolveD`, which randomly selects agents from the top-3 sets each round, performs slightly worse than the fixed variant `TUMIX-Evolve` across all three benchmarks. Agent quality may decline because useful specialized agents get replaced, reducing their ability to interpret or reflect on others' answers. However, the effect is minimal, so we conclude that agent evolution has no meaningful impact.

**Impact of number of agent types**   We next examine the marginal benefit of increasing the number of agents in `TUMIX`. Agents are randomly sampled from the pool of 30, with each contributing one inference per round. To isolate this effect, we exclude termination and selection and report the evolution of peak average accuracy across rounds. As shown in Fig. 10, accuracy rises quickly when the number of agents is below 12, but gains become negligible thereafter. This indicates that beyond a certain point, increasing agent types and inference budget yields little benefit. Under a constrained set of tools, the marginal gains from additional agent types diminish to nearly zero because agent diversity and viable tool-use strategies are inherently limited, while the growing number of candidate answers makes round-by-round selection increasingly challenging. Based on these results, we decide to only include 15 agents in `TUMIX` to balance performance and cost.

## 5.4 SCALING CURVES: PERFORMANCE VS. COSTS

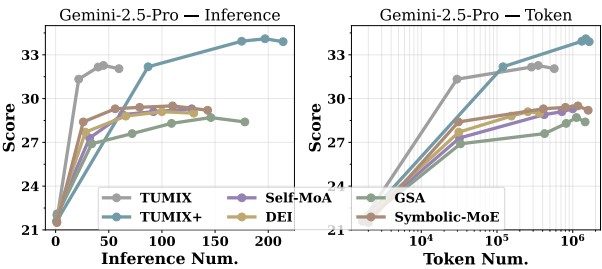

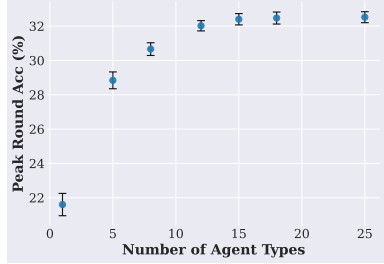

Figure 9: Scaling behavior of HLE scores relative to inference cost and total token count across different tool-augmented test-time scaling methods, where the token count includes both input and output tokens.

Figure 10: Peak round accuracy versus number of agent types. Agent types randomly sampled from the 30 well-performing agents with three repetition. Each agent infers once per round.

We compare the scaling behavior of different tool-augmented test-time scaling methods in terms of inference and token costs. In `TUMIX` and `Self-MoA`, scaling comes from adding more refinement rounds; in `GSA`, `DEI`, and `Symbolic-MoE`, from repeating inference; and in `TUMIX+`, from both. As shown in Fig. 9 and Appendix Fig. 12, `TUMIX` consistently outperforms other methods, achieving the highest scores with fewer inference steps and tokens. `TUMIX+` pushes peak performance further by repeating inference four times across the first two refinement rounds, but at substantially higher cost and lower efficiency. In Appendix Section G, we report tool latency, code execution time, and API cost, showing that all tool operations are relatively negligible compared to LLM inference time and that `TUMIX` maintains lower or comparable financial costs than baselines. Overall, test-time scaling demands far more inferences and roughly two orders of magnitude more tokens, a seemingly unavoidable trade-off.

## 6 CONCLUSION

We introduce Tool-use Mixture (`TUMIX`), a framework that leverages diverse tool-use strategies to improve reasoning in LLMs. By coordinating multiple agents with complementary approaches to textual reasoning, coding, and search, `TUMIX` substantially improves performance across challenging benchmarks, including HLE, GPQA, and AIME. Our findings highlight that diversity and quality of agents, rather than scale alone, drive these gains. Furthermore, automatic generation of agents and principled termination strategies enable both higher accuracy and significant efficiency improvements, reducing inference cost by nearly half without sacrificing performance. We reveal that structured diversity and selective refinement are key to maximizing the potential of tool-augmented LLMs.

## ETHICS STATEMENT

This paper contributes to advancing Foundation Models by augmenting language models with Code Interpreter and Search tools via test-time scaling, which has strong potential to improve performance and alignment with human preferences. However, such capabilities are inherently dual-use, the same techniques that augment models toward harmless outputs can, with minor changes, be misused to generate harmful content. While misuse is a concern, we believe the broader societal benefits outweigh the risks.

## REPRODUCIBILITY STATEMENT

For better reproducibility, we include detailed descriptions of 15 pre-designed agents and 15 LLM-generated agents in Table 1 and Table 22. The prompts of all agents are in Appendix Sec. C. The complete algorithm of TUMIX is illustrated in Appendix Sec. B. Our code and dataset will be made publicly available under an open-source license following the acceptance of the paper.

## LARGE LANGUAGE MODEL USAGE FOR WRITING

In this paper, we use LLMs—specifically Gemini and ChatGPT—as general-purpose writing aids. Draft text was provided to these models for grammatical correction and structural refinement, after which the output was verified and further edited when necessary. Their use was strictly limited to text refinement; they were not employed to generate new content or references.

## ACKNOWLEDGMENTS

The authors would like to thank Zifeng Wang for the helpful discussion and comments on the work. The work was also partly supported by ONR under Award N00014-22-1-2478 and MIT-IBM Watson AI Lab. This article solely reflects the conclusions of its authors and not the sponsors.

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

## Appendix–TUMIX: Multi-Agent Test-Time Scaling with Tool-Use Mixture

## A  EXAMPLE OF GPT-5 FAILURE IN CODE/TEXT DECISION

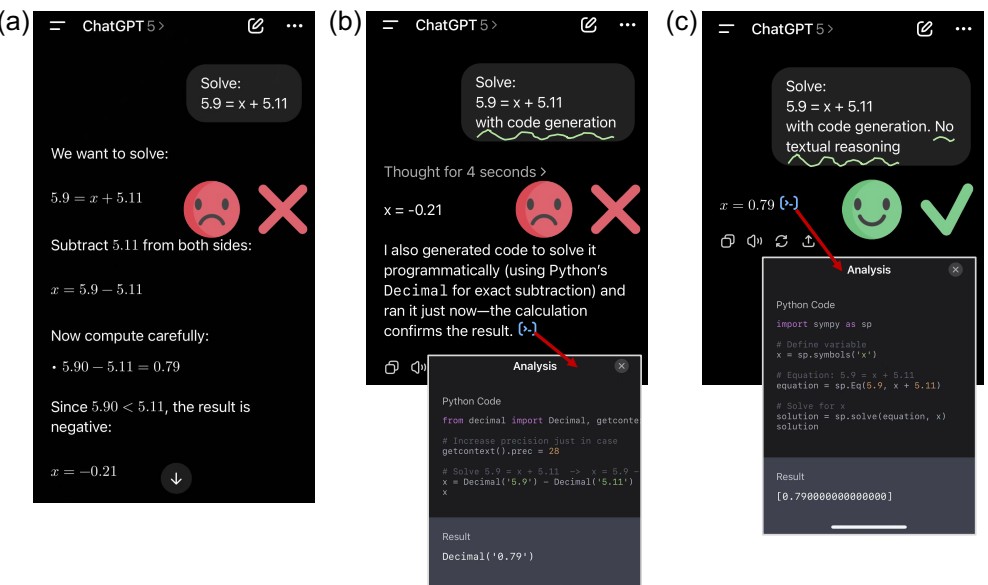

Figure 11: Example of GPT-5 failure in code/text decision. In this case, the question is incorrectly solved with textual reasoning (a) but can be easily addressed through code generation (c). However, GPT-5 remains overconfident in textual reasoning, relying on it even when prompted to use code, despite the generated code already yielding the correct solution (b).

# B  ALGORITHM OF TUMIX

---

**Algorithm 1** TUMIX: Multi-Agent Test-Time Scaling (answers only)

---

**Require:** question $q$; agent pool $\mathcal{S} = \{s_1, \ldots, s_K\}$  $\triangleright$ 15 pre-designed agents (Code / Search / Dual-Tool variants)

**Require:** minimum rounds $r_{\min} = 2$; maximum rounds $r_{\max}$; tool-interaction budget $R_{\text{tool}} = 5$; code time limit $\tau = 60s$

1: $\mathcal{A}_0 \leftarrow \varnothing$  $\triangleright$ answers from prior rounds
2: **for** $r = 1, 2, \ldots, r_{\max}$ **do**
   // **Round-$r$ message passing: each agent reads $q$ + prior answers**
3:     **parallel for** each $s \in \mathcal{S}$
4:         $p_r^s \leftarrow$ BUILDPROMPT$(q, \mathcal{A}_{r-1})$  $\triangleright$ concatenate $q$ with all answers from prior round
5:         $y_r^s \leftarrow$ AGENTCALL$(s, p_r^s, R_{\text{tool}}, \tau)$  $\triangleright$ tool-augmented reasoning (Alg. 2)
6:         $\mathcal{A}_r \leftarrow \{y_r^s : s \in \mathcal{S}\}$
7:         **if** $r \geq r_{\min}$ **and** LLMTERMINATE$(q, \mathcal{A}_r) =$ STOP **then**
8:             **break**
9:         **end if**
10: **end for**
11: $a^\star \leftarrow$ MAJORITYVOTE$(\mathcal{A}_r)$
12: **return** $a^\star$

---

**Algorithm 2** AGENTCALL$(s, p, R_{\text{tool}}, \tau)$: tool-augmented reasoning for agent $s$ (returns answer only)

---

1: $h \leftarrow p; b \leftarrow 0$  $\triangleright$ $h$ is the running context; $b$ counts tool interactions
2: **while** $b < R_{\text{tool}}$ **do**
3:     $o \leftarrow$ LLMGENERATE$(s, h)$  $\triangleright$ run agent $s$ with its strategy/prompt
4:     **if** $o$ contains a final answer $y$ **then**
5:         **return** $o$
6:     **else if** $o$ proposes code and $s$ allows `Code Interpreter` **then**
7:         $r \leftarrow$ EXECUTECODE$(o, \tau)$  $\triangleright$ hard limit $\tau = 60s$; capture stdout/plots/files
8:         **if** RUNTIMEERROR$(r)$ **then**
9:             $h \leftarrow h \parallel$ "Runtime error:" $r$; $b \leftarrow b + 1$; **continue**
10:        **else**
11:            $h \leftarrow h \parallel$ "Code result:" $r$; $b \leftarrow b + 1$; **continue**
12:        **end if**
13:    **else if** $o$ issues a search query and $s$ allows `Search` **then**
14:        $E \leftarrow$ WEBSEARCH$(o)$  $\triangleright$ supports `gs`/`llm`/`com` variants
15:        $h \leftarrow h \parallel$ "Retrieved evidence:" $E$; $b \leftarrow b + 1$; **continue**
16:    **else**
17:        $h \leftarrow h \parallel$ "Continue reasoning with current context."  $\triangleright$ encourage self-reflection
18:    **end if**
19: **end while**
20: $o \leftarrow$ LLMGENERATE$(s, h)$  $\triangleright$ budget exhausted; force a decision
21: **return** $o$

---

# C PROMPTS OF TUMIX

Table 3: Prompt for answer refinement based on all agent answers in the previous round.

---

**Task**: Decide the final answer based on the following answers from other agents.

**Question**:

{question}

**Candidate answers from several methods**:

{joined_answers}

Based on the candidates above, analyze the question step by step and try to list all the careful points. In the end of your response, directly output the answer to the question with the format <<<answer content>>>.

---

Table 4: Prompt for LLM-as-Judge of refinement termination.

---

**Task**: Carefully assess whether the answers below (enclosed by <<< >>>) show clear and strong consensus, or if another round of reasoning is needed to improve alignment.

**IMPORTANT**: If there are any differences in reasoning, phrasing, emphasis, conclusions, or interpretation of key details, you should conservatively decide to continue refinement.

The current round number is {round_num}. Note: **Finalizing before round 3 is uncommon and discouraged unless answers are fully aligned in both logic and language.**

**Question**:

{question}

**Candidate answers from different methods**:

{joined_answers}

**Instructions**:

1. Identify any differences in wording, structure, or logic.

2. Be especially cautious about subtle variations in conclusion or emphasis.

3. Err on the side of caution: if there's any ambiguity or divergence, recommend another round.

Output your reasoning first, then conclude clearly with <<<YES>>> if the answers are highly consistent and finalization is safe, or <<<NO>>> if further refinement is needed.

---

Table 5: Prompt for Gemini-2.5-Pro to synthesize diverse and high-quality agents based on the code of human pre-designed agents.

---

**Task**: Generate new, diverse, and high-quality agents based on the full code of existing agents. You have full access to tools Code Interpreter and Search. Output the complete implementation code for each new agent. Separate the agent framework code and the prompt code into two files.

**Instruction**: *Generate agents that are totally different in reasoning and tool-use strategy, not just limited to prompt optimization.*

**Existing agent codes**:

{agent code}

**Output**:

---

Table 6: Head prompt for `CoT Agent`.

- Analyze the question step by step and try to list all the careful points.
- Then try to acquire the final answer with step by step analysis.
- In the end of your response, directly output the answer to the question.

**Do not output the code for execution.**

Table 7: Head prompt for `CoT-Code Agent`.

You are a helpful AI assistant. Solve tasks using your coding skills.

In the following cases, suggest python code (in a python coding block) for the user to execute.

- Don't include multiple code blocks in one response, **only include one** in the response.
- Do not ask users to copy and paste the result. Instead, use the 'print' function for the output when relevant.

Think the task step by step if you need to. If a plan is not provided, explain your plan first. You can first output your thinking steps with texts and then the final python code.

**Remember in the final code you still need to output each number or choice in the final print!**

Start the python block with ```python

Table 8: Head prompt for `Code Agent`.

The User asks a question, and you solve it. You first generate the reasoning and thinking process and then provide the User with the final answer. During the thinking process, **you can generate python code** for efficient searching, optimization, and computing with the format of starting the python block with ```python. **A code query must involve only a single script that uses 'print' function for the output.**. Once the code script is complete, stop the generation. Then, the code interpreter platform will execute the code and return the execution output and error. Once you feel you are ready for the final answer, directly return the answer with the format <<<answer content>>> at the end of your response. Otherwise, you can continue your reasoning process and possibly generate more code query to solve the problem.

# D    BASELINE METHODS

Table 9: Head prompt for `Dual-Tool Agent`.

The User asks a question, and you solve it. You first generate the reasoning and thinking process and then provide the User with the final answer.

During the thinking process, you can generate python code for efficient searching, optimization, and computing with the format of starting the python block with ` ```python `. `**A code query must involve only a single script that uses 'print' function for the output.**`.. Once the code script is complete, stop the generation. Then, the code interpreter platform will execute the code and return the execution output and error.

If you lack the related knowledge, you can use the Google Search Tool to search the web and get the information. You can call a search query with the format of `<search>your search query</search>`, e.g., `<search>Who is the current president of US?</search>`. The searched results will be returned between `<information>` and `</information>`. Once the search query is complete, stop the generation. Then, the search platform will return the searched results.

If you need to search the web, **do not generate code in the same response. Vice versa**. You can also solve the question without code and searching, just by your textual reasoning.

Once you feel you are ready for the final answer, directly return the answer with the format `<<<answer content>>>` at the end of your response. Otherwise, you can continue your reasoning process and possibly generate more code or search queries to solve the problem.

Table 10: Head prompt for `Guided Agent`.

You are guiding another TaskLLM to solve a task. You will be presented with a task that can be solved using textual reasoning, coding, and web searching. Sometimes the TaskLLM may need extra help to solve the task, such as generating code or searching the web. Then must follow the rules below for both query and return answer:

During the thinking process, you can generate python code for efficient searching, optimization, and computing with the format of starting the python block with ` ```python `. `A code query must involve only a single script that uses 'print' function for the output.`. Once the code script is complete, stop the generation. Then, the code interpreter platform will execute the code and return the execution output and error.

If you lack the related knowledge, you can use the Google Search Tool to search the web and get the information. You can call a search query with the format of `<search>your search query</search>`, e.g., `<search>Who is the current president of US?</search>`. The searched results will be returned between `<information>` and `</information>`. Once the search query is complete, stop the generation. Then, the search platform will return the searched results.

If you need to search the web, **do not generate code in the same response. Vice versa**. You can also solve the question without code and searching, just by your textual reasoning.

Once you feel you are ready for the final answer, directly return the answer with the format `<<<answer content>>>` at the end of your response. Otherwise, you can continue your reasoning process and possibly generate more code or search queries to solve the problem.

**Your goal is to determine which method will be most effective for solving the task.** Then you generate the guidance prompt for the TaskLLM to follow in the next round. The final returned guidance prompt should be included between `<<<` and `>>>`, such as `<<<You need to generate more complex code to solve...>>>`.

Now, here is the task:

Table 11: Baseline methods compared against `TUMIX`.

| Method Handle | Type | Description |
|---|---|---|
| `Majority-Vote` | Voting | A single agent runs multiple parallel inferences, with the final answer decided by majority voting, without sharing intermediate results. Uses `CS` agent. |
| `GSA` | Aggregation | Similar to `Majority-Vote`, but the same LLM generates a new response conditioned on multiple samples. Uses `CS` agent. |
| `Self-Reflection` | Iterative Refinement | A single agent iteratively refines its answer by reflecting on past responses (up to 10 accessible per round; varied to 8 or 15 in experiments, with no performance difference). Uses `CS` agent. |
| `SETS` | Multi-Trial Voting | The same LLM performs multiple self-reflection trials, and the final answer is chosen by majority vote. Uses `CS` agent. |
| `Self-MoA` | Best-Agent Selection | Selects the best-performing agent among 15 candidates for parallel sampling, answer sharing, and multi-round refinement. Adapted to select the best agent within the same LLM instead of the original setting, selecting the best LLM across different LLMs. |
| `Symbolic-MoE` | Expert Selection | Categorizes questions (e.g., algebra, probability, coding, biology), pre-tests the top 3 agents per category, and LLM judges the question category and assigns test questions to these top agents for sampling and aggregation. |
| `DEI` | Committee Heuristic | Selects the top 5 agents, generates multiple answers via repetitive sampling, and then uses a predefined agent committee with heuristics to select the best answer. |
| `SciMaster` | Critic and Refine | Samples the same pre-designed tool-use agent five times, then employs other agents to critique, refine, and aggregate the answers. Original prompts/agents retained. |

Table 12: Experimental results of baseline and proposed methods on HLE, GPQA, and AIME 24&25. Except for the single-inference `w/o TTS` and the scaled-up `TUMIX+`, all methods use comparable inference costs for scaling. For some methods, Gemini-2.5-Pro's HLE results are used to select agents within their agentic framework. In these cases, the method has prior knowledge of HLE and the results cannot be strictly regarded as test performance. Such cases are marked with $^*$ in the HLE results. All the values are the average of three repetitive runs. Here we report both the mean values and standard deviations, covering around 68% confidence intervals.

| | | BASELINE METHODS | | | | | | PROPOSED METHODS | |
|---|---|---|---|---|---|---|---|---|---|
| ACCURACY % | W/O TTS | SELF-MOA | SYMBOLIC-MOE | DEI | SCIMASTER | GSA | TUMIX | TUMIX+ |
| | | | | **GEMINI-2.5-PRO** | | | | | |
| HLE | $21.6 \pm 0.3$ | $29.3^* \pm 0.5$ | $29.5^* \pm 0.4$ | $29.1^* \pm 0.6$ | $26.9 \pm 0.4$ | $28.7 \pm 0.7$ | $32.3 \pm 0.4$ | $34.1 \pm 0.5$ |
| GPQA | $84.6 \pm 0.5$ | $85.5 \pm 0.4$ | $86.7 \pm 0.5$ | $86.0 \pm 0.6$ | $86.9 \pm 0.6$ | $85.8 \pm 0.6$ | $87.9 \pm 0.4$ | $88.3 \pm 0.6$ |
| AIME | $87.3 \pm 0.4$ | $94.7 \pm 0.5$ | $94.7 \pm 0.6$ | $95.0 \pm 0.4$ | $94.1 \pm 0.6$ | $93.7 \pm 0.4$ | $96.7 \pm 0.5$ | $96.7 \pm 0.4$ |
| **AVE.** | $\mathbf{64.5 \pm 0.4}$ | $\mathbf{69.8 \pm 0.5}$ | $\mathbf{70.3 \pm 0.5}$ | $\mathbf{70.0 \pm 0.5}$ | $\mathbf{69.3 \pm 0.5}$ | $\mathbf{69.4 \pm 0.6}$ | $\mathbf{72.3 \pm 0.4}$ | $\mathbf{73.0 \pm 0.5}$ |
| | | | | **GEMINI-2.5-FLASH** | | | | | |
| HLE | $9.7 \pm 0.5$ | $18.2 \pm 0.6$ | $18.5 \pm 0.7$ | $19.3 \pm 0.8$ | $18.0 \pm 0.6$ | $17.4 \pm 0.5$ | $21.2 \pm 0.5$ | $23.1 \pm 0.7$ |
| GPQA | $50.0 \pm 0.5$ | $65.4 \pm 0.6$ | $64.1 \pm 0.7$ | $64.9 \pm 0.7$ | $67.9 \pm 0.7$ | $62.6 \pm 0.7$ | $77.3 \pm 0.7$ | $82.1 \pm 0.6$ |
| AIME | $70.0 \pm 0.7$ | $80.3 \pm 0.7$ | $80.7 \pm 0.8$ | $82.3 \pm 0.6$ | $79.1 \pm 0.6$ | $79.7 \pm 0.6$ | $83.3 \pm 0.7$ | $86.7 \pm 0.7$ |
| **AVE.** | $\mathbf{43.2 \pm 0.6}$ | $\mathbf{54.7 \pm 0.6}$ | $\mathbf{54.4 \pm 0.7}$ | $\mathbf{55.5 \pm 0.7}$ | $\mathbf{55.0 \pm 0.6}$ | $\mathbf{53.2 \pm 0.6}$ | $\mathbf{60.6 \pm 0.6}$ | $\mathbf{64.0 \pm 0.7}$ |

# E  STATISTICAL SIGNIFICANCE OF TESTED RESULTS ON TUMIX AND BASELINE METHODS.

**Paired significance testing.**  As for the method to calculate p-value, we perform a two-tailed paired $t$-test on their run-wise score differences $d_i = x_i - y_i$. The test statistic is

$$t = \frac{\bar{d}}{s_d/\sqrt{n}}, \qquad s_d = \sqrt{\frac{\sum_i (d_i - \bar{d})^2}{n-1}},$$

with $\mathrm{df} = n - 1$ degrees of freedom. The corresponding $p$-value is

$$p = 2\big(1 - F_t(|t|; \mathrm{df} = n - 1)\big),$$

where $F_t$ is the cumulative distribution function of the $t$ distribution. We consider differences statistically significant when $p < 0.05$.

Table 13: Statistical significance of **TUMIX** vs. baselines (**DEI**, **SciMaster**, **Symbolic-MoE**, **GSA**, **Self-MoA**) on each dataset. P-values come from two-tailed paired $t$-tests computed over run-wise score differences ($d_i = x_i - y_i$) across $n$=3 runs. A ✓ indicates $p$<0.05 (statistically significant), while ○ denotes non-significant differences.

| Task | TUMIX | vs. DEI | | vs. SciMaster | | vs. Symbolic-MoE | | vs. GSA | | vs. Self-MoA | |
|---|---|---|---|---|---|---|---|---|---|---|---|
| | | $p$-value | Sig. | $p$-value | Sig. | $p$-value | Sig. | $p$-value | Sig. | $p$-value | Sig. |
| *Gemini-2.5-Pro* | | | | | | | | | | | |
| HLE | $32.3 \pm 0.4$ | $< 0.01$ | ✓ | $< 10^{-4}$ | ✓ | $< 0.01$ | ✓ | $< 0.01$ | ✓ | $< 0.01$ | ✓ |
| GPQA | $87.9 \pm 0.4$ | 0.014 | ✓ | 0.084 | ○ | 0.034 | ✓ | 0.010 | ✓ | 0.002 | ✓ |
| AIME | $96.7 \pm 0.5$ | 0.011 | ✓ | $< 0.01$ | ✓ | 0.012 | ✓ | 0.002 | ✓ | 0.008 | ✓ |
| *Gemini-2.5-Flash* | | | | | | | | | | | |
| HLE | $21.2 \pm 0.5$ | 0.033 | ✓ | $< 0.01$ | ✓ | $< 0.01$ | ✓ | $< 0.01$ | ✓ | $< 0.01$ | ✓ |
| GPQA | $77.3 \pm 0.7$ | $< 10^{-4}$ | ✓ | $< 10^{-4}$ | ✓ | $< 10^{-4}$ | ✓ | $< 10^{-4}$ | ✓ | $< 10^{-4}$ | ✓ |
| AIME | $83.3 \pm 0.7$ | 0.135 | ○ | $< 0.01$ | ✓ | 0.014 | ✓ | 0.003 | ✓ | 0.006 | ✓ |

## F  Impacts of minimum refinement round on TUMIX performance and efficiency.

To mitigate premature termination in our semi-adaptive refinement process, we enforce a minimum number of refinement rounds. Table 14 presents an ablation study evaluating this hyperparameter. We observe that enforcing at least two refinement rounds substantially improves performance across both Gemini-2.5-Pro and Gemini-2.5-Flash models. Increasing the minimum round number beyond two provides negligible gains in HLE scores but incurs a steep rise in token consumption. Hence, we adopt a minimum of two rounds as the default setting, achieving the best trade-off between performance and computational efficiency.

Table 14: Ablation study on the minimum refinement round requirement in our semi-adaptive termination mechanism. Setting the minimum round number to 2 consistently improves HLE scores across models while maintaining a favorable token cost. Increasing the round number beyond 2 yields marginal gains at substantially higher computational costs.

| Minimum Round Number | 1 | 2 | 3 | 4 |
|---|---|---|---|---|
| HLE Score (Gemini-2.5-Pro) | 31.3 | **32.2** | 32.3 | 32.1 |
| Token Cost (Gemini-2.5-Pro, ×1k tokens) | 29.6 | **285** | 350 | 570 |
| HLE Score (Gemini-2.5-Flash) | 21.7 | **23.0** | 23.1 | 23.0 |
| Token Cost (Gemini-2.5-Flash, ×1k tokens) | 22.7 | **230** | 300 | 522 |

## G  Analysis of runtime stability and financial efficiency.

Regarding practical runtime, we opted for a more stable and reproducible metric. Wall-clock time can be highly volatile and difficult to compare fairly, as it is affected by factors such as network latency, server load, and specific hardware conditions. To ensure a standardized and hardware-agnostic comparison, the original paper reports API token counts and LLM inference numbers as direct proxies for computational cost. This approach allows our evaluation to reflect the intrinsic efficiency of the methods themselves, independent of experimental environments.

Although overall wall-clock time is not a reliable performance metric, in Table 15 we additionally report the average code execution and search latency. While these times are influenced by the content of generated code and search queries, we find that the average latencies are very similar across TUMIX and baseline methods. The average runtime per query for all tools remains under 7 seconds, which is negligible compared to LLM inference time (typically over 50 seconds for Gemini-2.5-Pro). Since each tool use corresponds to one LLM inference, this confirms that tool-use latency is not a computational bottleneck.

Table 15: Average tool latency per query. While tool latency varies slightly by task, all remain well below the typical LLM inference time, indicating that tool invocation overhead is negligible.

| Tools | Average Runtime (seconds) |
|---|---|
| Code Interpreter Execution | 2.3 |
| Google Search API | 1.2 |
| Gemini-2.5-Pro Search | 6.9 |
| Gemini-2.5-Flash Search | 4.5 |

We also report the average financial API cost per sample (including input/output tokens, Google Search API, and Gemini Search API) for TUMIX and strong baseline methods in Table 16. As shown, TUMIX incurs lower or comparable costs relative to most baseline methods, despite achieving higher performance and efficiency.

These analyses demonstrate that TUMIX achieves strong computational and financial efficiency while maintaining superior performance compared to baseline methods. The inclusion of latency and cost analyses further supports the practicality of our approach.

Table 16: Average financial API cost per sample in HLE/GPQA/AIME (in USD). TUMIX achieves competitive or lower costs than baseline methods while maintaining superior performance and computational efficiency.

| Method | Gemini-2.5-Pro | Gemini-2.5-Flash |
|---|---|---|
| **TUMIX** | **1.70** | **0.82** |
| DEI | 1.59 | 0.84 |
| SciMaster | 1.93 | 0.92 |
| Symbolic-MoE | 2.19 | 0.99 |
| GSA | 2.25 | 0.99 |
| Self-MoA | 2.23 | 0.96 |

## H    EXPERIMENTAL RESULTS OF `w/o TTS`, `DEI`, `TUMIX` WITH FOUR DIFFERENT BASE MODELS.

Table 17: Experimental results of `w/o TTS`, `DEI`, `TUMIX` with four different base models.

| Model | HLE | GPQA | AIME 2024&2025 |
|---|---|---|---|
| Claude-sonnet-4-20250514 | 8.2, 15.8, **21.8** | 44.6, 61.4, **72.3** | 34.4, 55.0, **70.8** |
| DeepSeek-R1 | 15.2, 23.7, **29.0** | 74.7, 78.0, **81.2** | 69.3, 85.0, **88.3** |
| GPT-oss-120B | 13.6, 15.4, **17.8** | 66.3, 70.1, **72.3** | 60.2, 82.7, **89.1** |
| Qwen3-32B | 13.1, 16.0, **18.0** | 54.6, 64.1, **68.3** | 59.6, 84.4, **88.1** |

# I PERFORMANCE OF TUMIX ON OPEN-ENDED SUMMARIZATION TASKS.

Table 18: Performance of TUMIX on open-ended summarization tasks from SCROLLS. TUMIX consistently outperforms both the baseline method without test-time scaling `w/o TTS` and the best baseline `DEI` across datasets and models, achieving the best ROUGE-1/2/L F1 scores.

| Metric | Model | w/o TTS | DEI (Zhang et al., 2024) | TUMIX |
|---|---|---|---|---|
| **ROUGE-1 (GovReport)** | Gemini-2.5-Flash | 0.440 | 0.458 | **0.466** |
| | GPT-oss-120B | 0.447 | 0.462 | **0.468** |
| **ROUGE-2 (GovReport)** | Gemini-2.5-Flash | 0.162 | 0.176 | **0.182** |
| | GPT-oss-120B | 0.152 | 0.169 | **0.175** |
| **ROUGE-L (GovReport)** | Gemini-2.5-Flash | 0.204 | 0.216 | **0.222** |
| | GPT-oss-120B | 0.203 | 0.214 | **0.220** |
| **ROUGE-1 (SummScreen-FD)** | Gemini-2.5-Flash | 0.282 | 0.315 | **0.323** |
| | GPT-oss-120B | 0.273 | 0.304 | **0.310** |
| **ROUGE-2 (SummScreen-FD)** | Gemini-2.5-Flash | 0.056 | **0.064** | 0.063 |
| | GPT-oss-120B | 0.051 | 0.058 | **0.060** |
| **ROUGE-L (SummScreen-FD)** | Gemini-2.5-Flash | 0.137 | 0.149 | **0.153** |
| | GPT-oss-120B | 0.134 | 0.146 | **0.149** |

## J  AGENT ACCURACY AND COVERAGE OVER MULTI-ROUND ANSWER REFINEMENT ON HLE

Table 19: Accuracy of each agent, average accuracy, and coverage across rounds for HLE. `Dual-Tool Agent`, `Guided Agent`, and `Guided Agent+` have three variants with different search strategies: Google Search API (`gs`), inherent LLM search function (`llm`), or their combination (`com`).

| Humanity's Last Exam (HLE) | RD 1 | RD 2 | RD 3 | RD 4 | RD 5 | RD 6 |
|---|---|---|---|---|---|---|
| **Coverage** | **51.92** | **44.20** | **43.48** | **37.04** | **34.85** | **33.87** |
| **Average** | **21.13** | **28.72** | **30.37** | **31.70** | **32.16** | **32.37** |
| w/o TTS | 20.32 | 28.08 | 30.88 | 31.60 | 32.18 | 32.36 |
| CoT Agent (CoT) | 20.84 | 28.16 | 30.40 | 31.12 | 32.30 | 32.48 |
| CoT-Code Agent ($CoT_{code}$) | 18.36 | 28.28 | 31.40 | 31.52 | 31.96 | 32.40 |
| Search Agent (S) | 21.72 | 29.04 | 28.84 | 32.08 | 32.08 | 32.20 |
| Code Agent (C) | 21.16 | 29.68 | 31.40 | 31.96 | 32.12 | 32.36 |
| Code Agent+ ($C^+$) | 22.96 | 29.00 | 31.44 | 31.88 | 32.20 | 32.40 |
| Dual-Tool Agent ($CS_{gs}$) | 22.96 | 28.60 | 30.84 | 31.72 | 32.24 | 32.36 |
| Dual-Tool Agent ($CS_{llm}$) | 21.36 | 28.36 | 31.28 | 31.20 | 32.48 | 32.32 |
| Dual-Tool Agent ($CS_{com}$) | 20.76 | 28.56 | 30.36 | 31.44 | 32.32 | 32.48 |
| Guided Agent ($CSG_{gs}$) | 22.04 | 28.72 | 29.96 | 32.24 | 32.00 | 31.96 |
| Guided Agent ($CSG_{llm}$) | 21.20 | 28.64 | 29.20 | 31.52 | 32.16 | 32.32 |
| Guided Agent ($CSG_{com}$) | 20.76 | 28.92 | 29.88 | 31.88 | 32.20 | 32.40 |
| Guided Agent+ ($CSG^+_{gs}$) | 20.56 | 29.32 | 30.36 | 31.92 | 31.84 | 32.52 |
| Guided Agent+ ($CSG^+_{llm}$) | 21.56 | 28.80 | 29.20 | 31.64 | 32.16 | 32.52 |
| Guided Agent+ ($CSG^+_{com}$) | 20.44 | 28.68 | 30.08 | 31.72 | 32.28 | 32.48 |

# K ILLUSTRATION AND EXPERIMENTAL RESULTS OF TUMIX VARIANTS

Table 20: TUMIX framework and its variants, designed to ablate core components.

| Method Handle | Component Ablated | Description |
|---|---|---|
| TUMIX | Main Method | (Default) Uses an LLM query to determine the optimal termination round (min. 2) and majority vote for final selection. |
| TUMIX-Rule | Termination | Replaces the LLM-query termination with a rule: stops when the majority answer stabilizes across two consecutive rounds. |
| TUMIX-Fixed | Termination | Replaces smart termination with a fixed 5-round limit, followed by majority voting for selection. |
| TUMIX-FixedR | Termination & Selection | Uses a fixed 5-round limit, followed by random selection. |
| TUMIX-Evolve | Agent Composition | Replaces the 15 human-designed agents with a static group of top-performing, LLM-generated agents for each refinement round. |
| TUMIX-EvolveD | Agent Composition | Extends the above by dynamically sampling a new agent group from the top-3 Evolved Agent groups for each refinement round. |
| TUMIX-Single | Agent Diversity | Ablates diversity by replacing the 15 distinct agents with a single agent type from the $CS$ family. |
| TUMIX-Three | Agent Diversity | Reduces diversity by using only three agent types ($CS$, $C^+$, and $CSO$), each sampled 5 times per round. |
| TUMIX+ | Inference Scaling | Extends 'TUMIX' with test-time scaling, running four inference passes per agent at different temperatures for the initial two rounds. |

Table 21: Experimental results of TUMIX variants. All the values are the average of three repetitive runs.

| METHODS 
 SUCCESS RATE % | TUMIX VARIANTS | | | | | | | | |
|---|---|---|---|---|---|---|---|---|---|
| | TUMIX | TUMIX-SINGLE | TUMIX-THREE | TUMIX-FIXEDR | TUMIX-FIXED | TUMIX-RULE | TUMIX-EVOLVE | TUMIX-EVOLVED | TUMIX+ |
| **GEMINI-2.5-PRO** | | | | | | | | | |
| HLE | 32.3 | 29.0 | 30.2 | 32.4 | 32.4 | 32.4 | 32.7 | 32.1 | 34.1 |
| GPQA | 87.9 | 86.1 | 86.6 | 86.8 | 86.7 | 87.7 | 88.1 | 87.4 | 88.3 |
| AIME 24&25 | 96.7 | 95.0 | 95.3 | 95.6 | 96.7 | 96.7 | 96.7 | 96.1 | 96.7 |
| **AVE. NORM.** | **72.3** | **70.0** | **70.7** | **71.6** | **71.9** | **72.3** | **72.5** | **71.9** | **73.0** |
| **GEMINI-2.5-FLASH** | | | | | | | | | |
| HLE | 21.2 | 18.2 | 18.6 | 20.9 | 20.8 | 21.3 | 21.9 | 21.3 | 23.1 |
| GPQA | 77.3 | 65.8 | 67.1 | 76.8 | 77.1 | 77.4 | 79.8 | 78.3 | 82.1 |
| AIME 24&25 | 83.3 | 80.6 | 81.2 | 83.3 | 83.3 | 83.3 | 86.7 | 86.7 | 86.7 |
| **AVE. NORM.** | **60.6** | **54.9** | **55.6** | **60.3** | **60.4** | **60.7** | **62.8** | **62.1** | **64.0** |

# L NEW AGENTS IN TUMIX COMPLETELY DESIGNED BY GEMINI-2.5-PRO AUTOMATICALLY

Table 22: Summary of 15 LLM-generated agents, categorized by their framework characteristics.

| Full Name | Short Name | Description |
|---|---|---|
| **Iterative Agents (Multi-turn conversational frameworks)** | | |
| Plan-Verify-Refine | PVR | Iteratively plans, executes one action (code or search), and refines based on checker feedback. |
| SearchThenCode | S→C | Enforces a search-first, then code execution sequence in an iterative loop. |
| CodeThenSearch | C→S | Enforces a code-first, then search execution sequence in an iterative loop. |
| ConstraintPrune-Solver | $CP_{solv}$ | Iteratively prunes the solution space using constraints, guided by a checker and tools (code/search). |
| MonteCarlo-Verify | MCV | Uses Monte Carlo sampling via code to find a likely answer and then deterministically verifies it. |
| Debate-CrossExam | DCE | Simulates a Proposer/Skeptic debate to guide tool use, with a checker for cross-examination. |
| MultiHop-Search-Aggregate | $S_m$→C | Enforces at least two sequential search actions before allowing any code execution. |
| TDD-Code-Solver | $TDD_{solv}$ | A TDD agent that lists tests, writes code to pass them, and uses a checker for iterative refinement. |
| **Sequential Agents (Few-shot, non-conversational frameworks)** | | |
| SearchThenAnswer | S→A | A two-step agent that mandates a single web search before formulating the final answer. |
| PlanThenCode | P→C | A two-step agent that first generates a plan, then a single code block to execute it. |
| VerifierRefine | VR | A three-step agent that generates a text answer, validates it with a checker, and then refines it. |
| ToolSelector | TS | Explicitly selects one tool (Search, Code, or Text) in the first step, then finalizes. |
| HypothesisPruner-Code | $HP_{code}$ | Generates code to enumerate and prune solution hypotheses based on problem constraints. |
| DualSearch-Consensus | $S^2_{con}$ | Issues two distinct search queries and then synthesizes the results into a consensus answer. |
| TDD-CodeThenFix | $TDD_{fix}$ | A Test-Driven Development approach that writes tests and code, then generates a fix if tests fail. |

Table 23: Comparison of original agent group and top-3 agent group used in `TUMIX`, each with 15 agents, either pre-designed or LLM-generated.

| Original | Top-3-1 | Top-3-2 | Top-3-3 |
|---|---|---|---|
| w/o TTS | HypothesisPruner-Code | TDD-Code-Solver | w/o TTS |
| CoT Agent | CoT Agent | CoT Agent | CoT Agent |
| CoT-Code Agent | Plan-Verify-Refine | CoT-Code Agent | CoT-Code Agent |
| Search Agent | Search Agent | Search Agent | SearchThenCode |
| Code Agent | Code Agent | Code Agent | TDD-Code-Solver |
| Code Agent+ | SearchThenCode | Code Agent+ | HypothesisPruner-Code |
| Dual-Tool Agent$_{gs}$ | Dual-Tool Agent$_{gs}$ | SearchThenCode | DualSearch-Consensus |
| Dual-Tool Agent$_{llm}$ | ConstraintPrune-Solver | Plan-Verify-Refine | MonteCarlo-Verify |
| Dual-Tool Agent$_{com}$ | MonteCarlo-Verify | Dual-Tool Agent$_{com}$ | ConstraintPrune-Solver |
| Guided Agent$_{gs}$ | Guided Agent$_{gs}$ | Guided Agent$_{gs}$ | Debate-CrossExam |
| Guided Agent$_{llm}$ | Guided Agent$_{llm}$ | Guided Agent$_{llm}$ | Guided Agent$_{llm}$ |
| Guided Agent$_{com}$ | Debate-CrossExam | Guided Agent$_{com}$ | Guided Agent$_{com}$ |
| Guided Agent+$_{gs}$ | Guided Agent+$_{gs}$ | MonteCarlo-Verify | Guided Agent+$_{gs}$ |
| Guided Agent+$_{llm}$ | SearchThenAnswer | Guided Agent+$_{llm}$ | Plan-Verify-Refine |
| Guided Agent+$_{com}$ | DualSearch-Consensus | DualSearch-Consensus | Guided Agent+$_{com}$ |

# M    SCALING BEHAVIOR OF GEMINI-2.5-FLASH

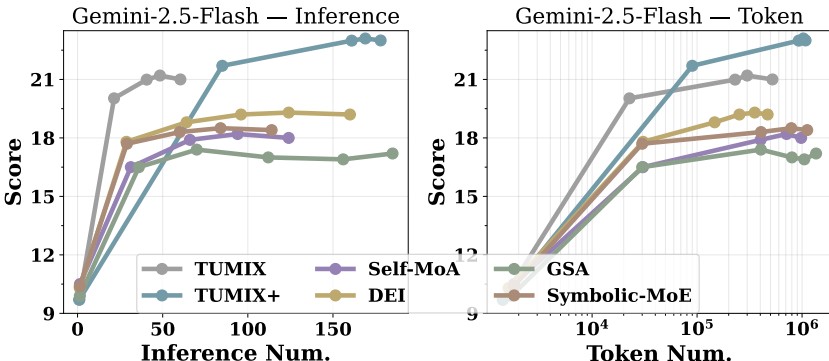

Figure 12: Scaling behavior of HLE scores in Gemini-2.5-flash relative to inference cost and total token count across different tool-augmented test-time scaling methods, where the token count includes both input and output tokens.

# N DIFFERENT TYPES OF LLMS AND MIXTURE OF LLMS TO ENHANCE PERFORMANCE

In Appendix Table 17, we extend our evaluation to four new base models: Claude-sonnet-4-20250514, DeepSeek-R1, GPT-oss-120B, and Qwen3-32B. We compare TUMIX with both the base method (w/o TTS) and the strongest baseline (DEI). Across all three benchmarks and four base models, TUMIX consistently yields higher performance under comparable token costs, demonstrating its robustness and broad applicability across heterogeneous LLMs. We also investigate heterogeneous mixtures of models, where agents are powered by different LLMs. Specifically, we evenly split agents between GPT-oss-120B and Qwen3-32B, which show similar capabilities in Appendix Table 17. As reported in Table 24, mixed-agent configurations outperform their single-model counterparts across rounds, confirming that heterogeneity enhances reasoning diversity and collaboration. However, when models differ substantially in capability (e.g., DeepSeek-R1 combined with weaker models), performance degrades, suggesting that mixtures are most effective when participating models have comparable strength. Overall, these findings reinforce that **diversity is critical**, and TUMIX generalizes well across both model families and heterogeneous agent settings.

Table 24: Mixed-LLM vs. Single-LLM agents. (**Top:**) mixtures of comparable LLMs (GPT-oss-120B and Qwen3-32B). (**Bottom:**) mixtures with a large capability gap (DeepSeek-R1 with GPT+Qwen).

| Base | Setting | HLE | GPQA | AIME 2024&2025 |
|------|---------|-----|------|----------------|
| *Comparable-strength models (GPT-oss-120B & Qwen3-32B)* | | | | |
| GPT-oss-120B | Single | 17.8 | 72.3 | 89.1 |
| GPT-oss-120B | **Mixed (GPT+Qwen)** | **19.8** | **75.5** | **90.6** |
| Qwen3-32B | Single | 18.0 | 68.3 | 88.1 |
| Qwen3-32B | **Mixed (GPT+Qwen)** | **19.0** | **70.0** | **91.4** |
| *Capability-gap mixture (DeepSeek-R1 with GPT+Qwen)* | | | | |
| DeepSeek-R1 | Single | **29.0** | **81.2** | 88.3 |
| DeepSeek-R1 | **Mixed (DeepSeek+GPT+Qwen)** | 22.6 | 73.4 | **91.8** |

# O TUMIX APPLICATION TO OPEN-ENDED TASKS: SUMMARIZATION

To further evaluate the generality of our approach beyond structured reasoning benchmarks, we test TUMIX on open-ended, real-world tasks where correctness may be ambiguous. We conduct experiments on long-document summarization, a representative open-ended task requiring both comprehension and abstraction. We adopt two benchmarks from the SCROLLS suite (Shaham et al., 2022): (1) *GovReport* (Huang et al., 2021), consisting of multi-page U.S. government reports (CRS/GAO), and (2) *SummScreen-FD* (Glaser et al., 2022), comprising TV-episode transcripts summarized into human-written recaps. These datasets feature realistic, lengthy documents with human references, enabling reproducible and objective evaluation. We report standard ROUGE-1/2/L F1 metrics (Lin, 2004), which measure unigram, bigram, and sequence-level overlap. As shown in Appendix Table 18, TUMIX consistently improves ROUGE-1/2/L scores compared to w/o TTS and DEI across both datasets using Gemini-2.5-Flash and GPT-oss-120B as base models, demonstrating the robust adaptability of TUMIX to open-ended, real-world tasks.

## P   LLM TOKEN CONFIDENCE OF GENERATED RESPONSES

**Confidence Distributions (Correct vs Wrong)**

Figure 13: Distribution of LLM response confidence for correct and wrong answers. The response confidence is calculated based on the average token probability of the whole generated response. Here we use the responses of agent `CoT` as representative, as we find the distribution characteristics are very close among different agents.

