# OpenReview forum: "TUMIX: Multi-Agent Test-Time Scaling with Tool-Use Mixture"
_ICLR.cc/2026/Conference — ICLR 2026 Poster_

### Official Review · Reviewer_tuT3 · 2025-10-27

**Soundness:** 3
**Presentation:** 3
**Contribution:** 2
**Rating:** 6
**Confidence:** 3

**Summary:**

This paper introduces TUMIX, an ensemble framework designed to solve the challenge of effectively combining LLM textual reasoning with Code Interpreter and Search tools. The framework operates by running multiple agents with diverse tool-use strategies in parallel, where agents iteratively refine their solutions by considering all outputs from the previous round. Experiments on Gemini-2.5-Pro and Flash show TUMIX significantly outperforms existing test-time scaling methods, achieving an average accuracy improvement of 3.55% over the best baseline.

**Strengths:**

1. The experiments show that TUMIX can beat several baselines across several highly challenging reasoning benchmarks (HLE, GPQA, AIME).
2. By using an LLM-as-Judge to determine the optimal stopping round, the framework achieves nearly identical performance to a fixed-round policy but reduces the inference cost.
3. The paper provides a key insight that, unlike in text-only scaling, agent diversity is crucial for tool-augmented reasoning and outperforms repeated sampling from a single strong agent.

**Weaknesses:**

1. The main results are averaged over only three repetitive runs. On complex benchmarks like GPQA and AIME, the variance in LLM reasoning can be high. The reported gains, while positive, are sometimes small (e.g., 96.7% vs. 94.7% on AIME-Pro; 87.9% vs. 86.9% on GPQA-Pro). With only three runs, it is difficult to ascertain if these differences are statistically significant or fall within the margin of error, which undermines the robustness of the SOTA claims. Additionally, the results omit the actual inference count and token metrics
2. The paper's reliance on "inference counts"  as a primary cost proxy is problematic; a single inference from a simple CoT agent is not "comparable" in cost to a single inference from a complex, multi-step dual-tool agent that consumes far more tokens.
3. The paper notes that performance gains plateau after ~12 agent types (Figure 10) and attributes this to the increasing difficulty of answer selection. However, this analysis fails to consider that all 30 agents (human- and LLM-designed) are fundamentally limited to the same two tools (Code and Search). The performance plateau may not be due to the number of agents, but rather the lack of tool diversity. See more details in Question.

**Questions:**

1. Regarding Agent Comparison (Figure 5): In the 3-agent vs. 15-agent comparison, the 3-agent group used three "strong agents" ($C^{+}$, $CS_{gs}$, and $CSG_{gs}$), each sampling 5 times. Were these three agents empirically determined to be the top-3 best-performing agents from the pool of 15?
2. The process for generating new agents (Section 5.3) is described as "query[ing] Gemini-2.5-Pro with current agent code examples". Could you provide more detail on this methodology? Specifically, what prompts were used to guide the LLM to generate strategies that "differ substantially from the original ones" (as seen in Table 14 ) rather than just minor prompt variations?
3. The framework was evaluated exclusively on Gemini-2.5-Pro and Gemini-2.5-Flash. Have you explored TUMIX's applicability to other models (e.g., GPT-5, Claude 4, or open-source models like DeepSeek and Qwen)? Furthermore, given the paper's emphasis on agent diversity, do you believe a heterogeneous mixture of models (i.e., agents powered by different LLMs) could be a viable strategy to further scale agent diversity and performance?
4. You correctly note that in later rounds, as answers converge, the choice of selection method (Majority Vote, LLM Selector, Random) becomes negligible. Does this imply that the primary benefit of the framework lies in the iterative refinement process itself rather than the final selection, and that for a highly-converged set of answers, a simple majority vote is sufficient?
5. You plot both inference number and token number against score in Figure 9. Given that token consumption is a more accurate measure of computational cost, and that different agents (e.g., CoT vs. Dual-Tool) have vastly different token profiles per "inference," would it be more transparent to frame the primary cost analysis around token count rather than inference count?

---

> ### Author Response · Authors · 2025-11-25
> **Response to Reviewer tuT3 (Q1- Part 1)**
>
> We sincerely thank the reviewer for the thoughtful evaluation of our work and for the detailed, inspiring questions and suggestions. We have **added the suggested experiments and analyses, and revised the paper accordingly, with the changed texts colored blue**. We hope the reviewer will help re-evaluate our work given this additional context.
>
> ---
>
> ### **Q1. Statistical significance, variance concerns, and missing inference/token metrics**
>
> **Reviewer’s concern:**
> On complex benchmarks like GPQA and AIME, the variance in LLM reasoning can be high. The reported gains, while positive, are sometimes small (e.g., 96.7% vs. 94.7% on AIME-Pro; 87.9% vs. 86.9% on GPQA-Pro). With only three runs, it is difficult to ascertain if these differences are statistically significant or fall within the margin of error, which undermines the robustness of the SOTA claims. Additionally, the results omit the actual inference count and token metrics.
>
> **Response:**
> Thank you for the thoughtful question. We would like to clarify a possible misunderstanding regarding the **reporting of inference counts and token metrics**. As noted in the **original paper (Figure 1, caption Line 47)**, **all methods were compared using approximately the same number of LLM inferences and tokens**. In addition, **Figure 9** presents **scaling curves** illustrating the relationship between **score vs. inference count** and **score vs. token count** for TUMIX and the baseline methods. Across both figures, **TUMIX shows consistently strong performance and scaling efficiency** compared to the baselines. Thus, we believe the experimental results **adequately account for** inference and token metrics.
>
> Thank you also for the helpful question regarding the **statistical significance** of TUMIX’s performance improvements. In the **original paper (Figure 1)**, we included **error bars (standard deviations)** alongside the mean performance of each method. Due to space constraints, these were not shown in **original paper Table 2**. In the **revised paper (Table 14 in Page 20, following Table 1)**, we now include both **mean and standard deviation values**.
>
> Overall, the **performance improvements of TUMIX** over the strongest baselines are **larger than the reported standard deviations**, indicating consistent and stable gains. Notably, TUMIX’s advantage is most apparent for the **smaller Gemini-2.5-Flash model**, demonstrating its robustness across model scales.
>
> In the **revised paper (Table 15 in Page 20, following Table 2)**, we additionally conduct **statistical significance tests** using *p*-values computed from repeated sampled scores. Across nearly all models and benchmarks, **p-values < 0.05**, confirming that TUMIX’s performance improvements are **statistically significant**.
>
> These results collectively demonstrate that the **improvements on smaller benchmarks (e.g., GPQA, AIME)** are **statistically valid** and not due to random variation.
>
> ---
>
> | Dataset | w/o TTS | Self-MoA | Symbolic-MoE | DEI | SciMaster | GSA | **TUMIX** | **TUMIX+** |
> |:--------|---------:|---------:|-------------:|----:|-----------:|----:|-----------:|------------:|
> | **Gemini-2.5-Pro** | | | | | | | | |
> | HLE | 21.6 ± 0.3 | 29.3 ± 0.5 | 29.5 ± 0.4 | 29.1 ± 0.6 | 26.9 ± 0.4 | 28.7 ± 0.7 | **32.3 ± 0.4** | **34.1 ± 0.5** |
> | GPQA | 84.6 ± 0.5 | 85.5 ± 0.4 | 86.7 ± 0.5 | 86.0 ± 0.6 | 86.9 ± 0.6 | 85.8 ± 0.6 | **87.9 ± 0.4** | **88.3 ± 0.6** |
> | AIME | 87.3 ± 0.4 | 94.7 ± 0.5 | 94.7 ± 0.6 | 95.0 ± 0.4 | 94.1 ± 0.6 | 93.7 ± 0.4 | **96.7 ± 0.5** | **96.7 ± 0.4** |
> | **Ave.** | 64.5 ± 0.4 | 69.8 ± 0.5 | 70.3 ± 0.5 | 70.0 ± 0.5 | 69.3 ± 0.5 | 69.4 ± 0.6 | **72.3 ± 0.4** | **73.0 ± 0.5** |
> | **Gemini-2.5-Flash** | | | | | | | | |
> | HLE | 9.7 ± 0.5 | 18.2 ± 0.6 | 18.5 ± 0.7 | 19.3 ± 0.8 | 18.0 ± 0.6 | 17.4 ± 0.5 | **21.2 ± 0.5** | **23.1 ± 0.7** |
> | GPQA | 50.0 ± 0.5 | 65.4 ± 0.6 | 64.1 ± 0.7 | 64.9 ± 0.7 | 67.9 ± 0.7 | 62.6 ± 0.7 | **77.3 ± 0.7** | **82.1 ± 0.6** |
> | AIME | 70.0 ± 0.7 | 80.3 ± 0.7 | 80.7 ± 0.8 | 82.3 ± 0.6 | 79.1 ± 0.6 | 79.7 ± 0.6 | **83.3 ± 0.7** | **86.7 ± 0.7** |
> | **Ave.** | 43.2 ± 0.6 | 54.7 ± 0.6 | 54.4 ± 0.7 | 55.5 ± 0.7 | 55.0 ± 0.6 | 53.2 ± 0.6 | **60.6 ± 0.6** | **64.0 ± 0.7** |
>
> - All values are averages of three runs. Reported standard deviations correspond to approximately **68% confidence intervals**.
>
> ---

---

> ### Author Response · Authors · 2025-11-25
> **Response to Reviewer tuT3 (Q1- Part 2, Q2)**
>
> ### Table: Statistical Significance of **TUMIX** vs. Baselines (**DEI**, **SciMaster**, **Symbolic-MoE**, **GSA**, **Self-MoA**)
>
> *p*-values are derived from **two-tailed paired *t*-tests** computed over run-wise score differences (\(d_i = x_i - y_i\)) across *n* = 3 runs.
> ✓ denotes *p* < 0.05 (statistically significant); ○ denotes non-significance.
>
> | **Task** | **TUMIX** | **vs. DEI** *p*-value | Sig. | **vs. SciMaster** *p*-value | Sig. | **vs. Symbolic-MoE** *p*-value | Sig. | **vs. GSA** *p*-value | Sig. | **vs. Self-MoA** *p*-value | Sig. |
> |:--|:--:|:--:|:--:|:--:|:--:|:--:|:--:|:--:|:--:|:--:|:--:|
> | **_Gemini-2.5-Pro_** ||||||||||||
> | HLE  | 32.3 ± 0.4 | < 0.01 | ✓ | < 10⁻⁴ | ✓ | < 0.01 | ✓ | < 0.01 | ✓ | < 0.01 | ✓ |
> | GPQA | 87.9 ± 0.4 | 0.014 | ✓ | 0.084 | ○ | 0.034 | ✓ | 0.010 | ✓ | 0.002 | ✓ |
> | AIME | 96.7 ± 0.5 | 0.011 | ✓ | < 0.01 | ✓ | 0.012 | ✓ | 0.002 | ✓ | 0.008 | ✓ |
> | **_Gemini-2.5-Flash_** ||||||||||||
> | HLE  | 21.2 ± 0.5 | 0.033 | ✓ | < 0.01 | ✓ | < 0.01 | ✓ | < 0.01 | ✓ | < 0.01 | ✓ |
> | GPQA | 77.3 ± 0.7 | < 10⁻⁴ | ✓ | < 10⁻⁴ | ✓ | < 10⁻⁴ | ✓ | < 10⁻⁴ | ✓ | < 10⁻⁴ | ✓ |
> | AIME | 83.3 ± 0.7 | 0.135 | ○ | < 0.01 | ✓ | 0.014 | ✓ | 0.003 | ✓ | 0.006 | ✓ |
>
> ---
>
> These analyses provide **strong statistical evidence** that **TUMIX’s performance gains**, including those on smaller benchmarks such as **GPQA** and **AIME**, are **statistically significant, consistent, and robust** across models and datasets.
>
> ---
>
> ### **Q2. Should cost analysis focus on token count rather than inference count?**
>
> **Reviewer’s concern:**
> The paper uses *inference count* as a primary cost proxy, but a single inference from a simple CoT agent is not comparable to a single inference from a complex multi-step dual-tool agent that consumes far more tokens. Since token consumption is a more accurate indicator of computational cost—and different agents have very different token profiles—would it be more transparent to center the primary cost analysis on *token count* instead of *inference count*?
>
> **Response:**
> Here, **inference** refers specifically to the **LLM inference numbers**, not the number of agent inferences. In this context, the **LLM inference count inherently reflects** the different query requirements between a **single-step CoT agent** and more **complex multi-step agents**.
>
> As the reviewer correctly notes, even for a single inference, a **one-step CoT agent** consumes **fewer tokens** than a **multi-step agent**, since the latter includes **longer context**, **reasoning history**, and **tool execution outputs**. This is precisely why we report the **total token cost** as an additional metric—it accounts for this **token imbalance** and provides a more **comprehensive and balanced measure** of computational cost.
>
> To ensure a **standardized and hardware-agnostic comparison**, the **original paper** reports both **API token counts** and **LLM inference numbers** as proxies for computational cost. For a more well-rounded cost analysis, we also report the **average financial API cost per sample (including input and output tokens, Google Search API, Gemini Search API)** for **TUMIX** and several strong baseline methods in the **revised paper (Table 18 in Page 22)**, reproduced below as **Table 1**. As shown, **TUMIX incurs lower or comparable costs** relative to most baselines, while achieving **higher performance and efficiency**.
>
> #### Table 2: Financial API cost per sample in HLE/GPQA/AIME (in USD)
> | **Method** | **Gemini-2.5-Pro** | **Gemini-2.5-Flash** |
> |:-------------|:-----------------:|:---------------------:|
> | **TUMIX** | **1.70** | **0.82** |
> | DEI | 1.59 | 0.84 |
> | SciMaster | 1.93 | 0.92 |
> | Symbolic-MoE | 2.19 | 0.99 |
> | GSA | 2.25 | 0.99 |
> | Self-MoA | 2.23 | 0.96 |
>
> ---
>
> These analyses demonstrate that **TUMIX** achieves **strong computational and financial efficiency**, while maintaining **superior performance** compared to baseline methods.
> We have added the above discussion and new experimental results in the revised paper Lines 483-485 and Appendix Section G.

---

> ### Author Response · Authors · 2025-11-25
> **Response to Reviewer tuT3 (Q3-Q5)**
>
> ### **Q3. Does the performance plateau arise from limited tool diversity rather than the number of agents?**
>
> **Reviewer’s concern:**
> Performance gains plateau after ~12 agent types (Figure 10), and the paper attributes this to increasing difficulty of answer selection. However, all 30 agents use only two tools (Code and Search). The plateau might therefore reflect **insufficient tool diversity**, not the number of agents.
>
> **Response:**
> Thank you for sharing your insightful opinion. We completely agree that the **diversity and performance of agent groups are inherently constrained by the limited diversity of available tools**, as our study currently focuses on only Code and Search. We expect that the **saturated agent number (around 12)** in our current experiments would likely increase if more effective and distinctive tools were available to the agents.
>
> In fact, this is one of the **key insights** we aim to highlight: under a **constrained set of tools**, increasing the number of agent types beyond a certain point no longer yields additional improvements, because the **marginal benefit from new tool-use strategies diminishes** to nearly zero, while the **complexity of answer selection becomes the dominant limiting factor**. We have added this clarification in the revised paper **Lines 457–460**.
>
> ---
>
> ### **Q4. Were the 3 agents in the 3-agent vs. 15-agent comparison the empirically top-performing ones?**
>
> **Reviewer’s concern:**
> In Figure 5, the 3-agent group uses three “strong agents” (\(C^{+}\), \(CS_{gs}\), \(CSG_{gs}\)). Were these empirically shown to be the **top-3 performers** among the 15-agent pool?
>
> **Response:**
> Yes, as shown in **Appendix Table 11 of the original paper**, the performance of each agent in the first round (without answer sharing) reflects the **individual capability of each agent** on the HLE benchmark. The results indicate that **$C^{+}$, $CS_{gs}$, and $CSG_{gs}$ achieve the top three scores** (22.96, 22.96, 22.04) among the 15 agents (average is 21.13).
>
> Based on this observation, we **select these three as the base agents** for the 3-agent method, as they are relatively stronger performers. This explanation has been added to the revised paper **Line 373 Page 7** for clarity.
>
> ---
>
> ### **Q5. How were new LLM-generated agents produced, and how was substantial strategy diversity ensured?**
>
> **Reviewer’s concern:**
> Section 5.3 states that new agents were generated by “querying Gemini-2.5-Pro with current agent code examples,” but provides little detail. What prompting strategy ensured **substantial differences** from existing agents instead of minor variations?
>
> **Response:**
> Thank you for the important question. The prompt used to query **Gemini-2.5-Pro** for designing new agents based on the **full code of all existing agents** is straightforward: we instruct the LLM to **generate more diverse and high-quality agents** and **output the full implementation code** for each. In the prompt, we instruct Gemini-2.5-pro to ‘generate agents that are totally different in reasoning and tool-use strategy, not just limited to prompt optimization’. We have added this prompt in the **revised paper Table 7 in Page 17** for clarity.
>
> The **agent selection strategy** is described in the **original paper (Page 8, Line 431)**. Specifically, we query **Gemini-2.5-Pro** to design **25 new agents**, evaluate their performance on **HLE** using **Gemini-2.5-Pro** as the base model, and retain the **15 best-performing** agents as the final set. The retained **top 15 agents** are combined with the **original 15 human-designed agents** to form new groups. These groups outperform the original human-designed agents on **HLE**, **GPQA**, and **AIME**, demonstrating strong generalizability.
>
> As shown in **Appendix Table 14 (Page 20)** of the original paper, the **LLM-generated agents** differ substantially in **strategies and frameworks**, extending well beyond prompt optimization. Many of these agents employ **unique tool-use strategies**, such as **forced switching**, **starting with code or search**, and **adaptive mode selection**. This increased **diversity of agent design** contributes directly to the **enhanced performance of TUMIX**.

---

> > ### Author Response · Authors · 2025-11-25
> > **Response to Reviewer tuT3 (Q6)**
> >
> > ### **Q6. Does TUMIX generalize to other models, and can heterogeneous mixtures of LLMs improve performance?**
> >
> > **Reviewer’s concern:**
> > The framework was evaluated only on Gemini-2.5-Pro/Flash. Has TUMIX been tested on **other frontier or open-source models** (e.g., GPT-5, Claude 4, DeepSeek, Qwen)? Given the emphasis on *agent diversity*, could **heterogeneous mixtures of models** serve as another axis to scale diversity and improve performance?
> >
> > **Response:**
> > Thank you for the constructive question. In the **revised paper (Table 3, Table 4, and Section 5.5)**, we include additional experiments using **four new base models**: *Claude-sonnet-4-20250514*, *DeepSeek-R1*, *GPT-oss-120B*, and *Qwen3-32B*. These models originate from **four different organizations**, with *DeepSeek-R1*, *GPT-oss-120B*, and *Qwen3-32B* being **smaller, open-source models**.
> >
> > We compare **TUMIX** against both the **base method without test-time scaling (w/o TTS)** and the **strongest baseline, DEI**. Since some models lack **built-in Search functions**, one Search agent is omitted in TUMIX, and the **three agents** originally relying on internal Search APIs are replaced with **Google Search APIs**. Consequently, the total number of agents decreases from **15 to 14**.
> >
> > As shown in **Table 1**, experimental results demonstrate that **TUMIX consistently improves performance** across all three benchmarks and all four base models. Under **comparable LLM token costs**, **TUMIX** also achieves **notably higher performance than DEI**, confirming its **effectiveness and general applicability**.
> >
> > #### Table 1: Performance across models (w/o TTS, DEI, TUMIX)
> > | **Model** | **HLE (w/o TTS, DEI, TUMIX)** | **GPQA (w/o TTS, DEI, TUMIX)** | **AIME 2024&2025 (w/o TTS, DEI, TUMIX)** |
> > |:-----------|:------------------------------:|:-------------------------------:|:-----------------------------------------:|
> > | Claude-sonnet-4-20250514 | 8.2, 15.8, **21.8** | 44.6, 61.4, **72.3** | 34.4, 55.0, **70.8** |
> > | DeepSeek-R1 | 15.2, 23.7, **29.0** | 74.7, 78.0, **81.2** | 69.3, 85.0, **88.3** |
> > | GPT-oss-120B | 13.6, 15.4, **17.8** | 66.3, 70.1, **72.3** | 60.2, 82.7, **89.1** |
> > | Qwen3-32B | 13.1, 16.0, **18.0** | 54.6, 64.1, **68.3** | 59.6, 84.4, **88.1** |
> >
> > ---
> >
> > Following the reviewer’s suggestion, we also tested **heterogeneous mixtures of models** (i.e., agents powered by different LLMs). We split the 14 agents equally between **GPT-oss-120B** and **Qwen3-32B** (7 each) and compared them to the single-base settings. These two models were chosen because their **individual performances are comparable**.
> >
> > The results show that both **mixed-agent configurations outperform their respective single-model baselines**, demonstrating that **heterogeneous model mixtures** can effectively enhance **diversity and reasoning performance** in multi-agent systems.
> >
> > #### Table 2: Mixed-LLM vs Single-LLM agents
> > | Base | Mixed or Single | **HLE** | **GPQA** | **AIME 2024&2025** |
> > |---|---|---:|---:|---:|
> > | **Comparable-strength models (GPT-oss-120B & Qwen3-32B)** |||||
> > | GPT-oss-120B | Single | 17.8 | 72.3 | 89.1 |
> > | GPT-oss-120B | **Mixed (GPT+Qwen)** | **19.8** | **75.5** | **90.6** |
> > | Qwen3-32B | Single | 18.0 | 68.3 | 88.1 |
> > | Qwen3-32B | **Mixed (GPT+Qwen)** | **19.0** | **70.0** | **91.4** |
> > | **Capability-gap mixture (DeepSeek-R1 with GPT+Qwen)** |||||
> > | DeepSeek-R1 | Single | **29.0** | **81.2** | 88.3 |
> > | DeepSeek-R1 | **Mixed (DeepSeek+GPT+Qwen)** | 22.6 | 73.4 | **91.8** |
> >
> > These results further reinforce our **conclusion** that **diversity is critical**, and they **demonstrate the strong generalizability of TUMIX** across both **different model types** and **heterogeneous model mixtures**.
> >
> >
> > Meanwhile, we also note that **mixtures of LLMs** bring additional benefits **only when the participating models have comparable performance**. In **Table 2**, we also mixed **DeepSeek-R1** with **GPT-oss-120B** and **Qwen3-32B**. However, we observed that this mixture led to **performance degradation** on HLE, as the weaker models (GPT-oss-120B and Qwen3-32B) reduced the overall group quality. This result suggests that **heterogeneous agent groups are most effective when the contributing models have similar capability levels**, as large performance disparities can negatively affect overall group reasoning quality.

---

> > > ### Author Response · Authors · 2025-11-25
> > > **Response to Reviewer tuT3 (Q7)**
> > >
> > > ### **Q7. Does convergence imply that iterative refinement—not the final selection policy—is the primary driver of performance?**
> > >
> > > **Reviewer’s concern:**
> > > In later rounds, answers become highly converged, making the choice among **Majority Vote**, **LLM Selector**, or **Random Selection** almost irrelevant. Does this mean that the main benefit of TUMIX comes from the **iterative refinement process**, and that once convergence is reached, a **simple majority vote** suffices?
> > >
> > > **Response:**
> > > Yes, the reviewer’s understanding correctly aligns with our results and conclusions. The observation that **cross-round sharing homogenizes answers** and that **later selection policies become near-equivalent** is an inherent characteristic of the TUMIX framework. In TUMIX, selection occurs through **iterative, round-by-round answer sharing and refinement among agents**, leveraging the LLM’s ability to evaluate and distinguish higher-quality answers through successive reasoning steps.
> > >
> > > Our experiments confirm that this mechanism is **highly effective**. Each refinement round brings **notable performance gains**, showing that the LLM can identify and converge toward better answers. Once the answers become highly converged, a **simple majority vote** is sufficient to determine the final answer, and further refinement is unnecessary.
> > >
> > > Therefore, TUMIX combines **LLM-based early termination** with **majority voting**, achieving up to a **49% reduction in token costs** while maintaining the same performance as full refinement. As presented in **Section 5.2 and Figure 7** of the original paper, this hybrid strategy allows TUMIX to preserve accuracy while **significantly improving efficiency**.
> > >
> > > ---
> > >
> > > **Thank you again for your efforts to help review our work and for the helpful suggestions. We look forward to discussing more with the reviewer for any further questions!**

---

### Official Review · Reviewer_Dy1M · 2025-11-01

**Soundness:** 4
**Presentation:** 4
**Contribution:** 3
**Rating:** 6
**Confidence:** 3

**Summary:**

This paper proposes TUMIX, a multi-agent framework for test-time scaling of tool-augmented LLMs. By orchestrating diverse specialized agents and enabling iterative response refinement, the system achieves an average gain of +3.55% on benchmarks including HLE and GPQA, using constrained inference budgets.

A key contribution of this work is the insight that agent diversity yields better results than homogeneous replication. Furthermore, the proposed adaptive termination mechanism reduces inference costs by approximately 49% while maintaining model accuracy. The experimental analysis on performance improvement in multi-agent systems also provides valuable reference for the research community.

**Strengths:**

- The TUMIX framework utilizes a coordinated multi-agent system in which specialized, tool-augmented agents perform parallel iterative refinement. Experiments report an average performance gain of 3.55% across several challenging benchmarks (HLE, GPQA, AIME 2024/2025), indicating the potential of this approach for test-time scaling.

- Through the introduction of a coverage rate metric and systematic ablation studies, the authors provide empirical support for the relationship between agent complementarity and overall task performance. This analytical methodology offers useful insights into the functioning of multi-agent ensembles.

- Empirical results consistently show that strategically diversifying agent composition leads to greater performance improvements than merely increasing the number of agents. This trend holds across various benchmark settings and model architectures.

- The proposed adaptive termination mechanism reduces computational costs by approximately 49%, while largely preserving model performance.

**Weaknesses:**

- Based on the experimental analysis presented in the paper, increasing the number of rounds generally leads to a decrease in agent coverage rate, suggesting reduced answer diversity and eventual performance convergence. However, in the GPQA experiment (Figure 5, Section 5.1), an increase in the number of rounds conversely results in a higher coverage rate. It would be interesting to understand the factors behind the contrasting behavior observed in the GPQA benchmark (Figure 5, Section 5.1).
- In Section 3.3, the authors note that many correct answers were incorrectly discarded by the LLM. Does this imply that the performance improvement is primarily influenced by the diversity of answers, as reflected by the coverage rate, while the use of the LLM to select responses may not contribute significantly to the performance gain?

**Questions:**

- Based on the experimental results in Figure 5, it appears that increasing the diversity of agent answers and mitigating the decline in coverage rate contribute to performance improvement. However, this observation seems to differ from the trends shown in Section 5.3 (Table 13). Some clarification regarding these differing outcomes would be helpful for a more comprehensive understanding.

---

> ### Author Response · Authors · 2025-11-25
> **Response to Reviewer Dy1M (Q1-Q2)**
>
> We sincerely thank the reviewer for the careful reading of our paper and the highly constructive questions and suggestions. We have incorporated the suggested analyses and added the corresponding discussions in the revised version. The **whole paper has been polished based on reviewers’ feedback, with the changed texts colored blue**. We hope the reviewer will help re-evaluate our work given this additional context.
>
> ---
>
> ### **Q1. Coverage increases on GPQA despite multi-round refinement normally reducing diversity**
>
> **Reviewer’s concern:**
> Figure 5 shows that in GPQA, coverage unexpectedly increases from round 4 to round 5, whereas in other benchmarks (HLE, AIME) coverage consistently decreases. What explains this contrasting behavior?
>
> **Response:**
> Thank you for the great observation and thoughtful question. Indeed, in Figure 5, the **increase in coverage accompanied by a decrease in average score** from round 4 to round 5 on the GPQA benchmark is an interesting phenomenon. We note that this behavior does **not occur in the HLE and AIME benchmarks**, where coverage continues to decrease and the average score steadily increases until reaching a plateau.
>
> Upon examining the answers from rounds 4 and 5 in GPQA across three repeated runs, we found that occasionally, **some agents discard the current common answer and propose a completely new one**, which can be correct or incorrect, but is most often wrong. This behavior explains the **temporary increase in coverage**, as some new correct answers are introduced, but also the **drop in average score**, since most new proposals mistakenly replace previously correct answers. We believe this opens up an interesting direction for future improvement.
>
> Such fluctuations are **common in iterative refinement processes**. When we extend the refinement to rounds 6 and 7, the coverage decreases again as expected, confirming that the observed anomaly is **transient**. We have included this discussion in the revised paper **Line 213 in Page 4** for completeness.
>
> ---
>
> ### **Q2. Is performance mainly driven by answer diversity rather than LLM-based selection?**
>
> **Reviewer’s concern:**
> Section 3.3 notes that many correct answers were incorrectly discarded. Does this imply that diversity (coverage) drives improvement more than LLM-based selection?
>
> **Response:**
> Thank you for the insightful question. Both **answer diversity** and **LLM-based answer selection** are essential and closely correlated factors. In general, test-time scaling methods rely on two key components:
>
> 1. **Generating multiple diverse thoughts or solutions**, and
> 2. **Selecting the best solution among them.**
>
> In TUMIX, selection occurs through **round-by-round answer sharing and refinement among agents**. This process leverages the LLM’s ability to evaluate and distinguish higher-quality answers through iterative reasoning. Our experiments show that this mechanism is effective, since the refinement round in TUMIX leads to **significant performance gains**, confirming that the LLM can indeed identify better answers.
>
> However, we also observe a noticeable gap between the **maximum coverage achieved in the first round** and the **final converged accuracy**, suggesting that the answer selection capability of LLMs still has room for improvement. Meanwhile, **answer diversity remains a crucial factor**. Higher diversity (i.e., broader coverage of reasoning paths) consistently leads to better final accuracy, likely because a more diverse set of answers provides richer reasoning insights that make selection easier and more reliable.
>
> In conclusion, **TUMIX enhances performance** by facilitating more effective answer selection through the generation of multiple **diverse** and **high-quality reasoning trajectories**.
>
> ---

---

> ### Author Response · Authors · 2025-11-25
> **Response to Reviewer Dy1M (Q3)**
>
> ### **Q3. Clarifying the relationship between diversity, coverage decline, and performance trends in Section 5.3**
>
> **Reviewer’s concern:**
> Figure 5 suggests that higher diversity and a slower decline in coverage lead to better performance. But Section 5.3 (Table 13) reports trends that appear different. Could the authors clarify?
>
> **Response:**
> The conclusion that **higher agent answer diversity** and a **slower decline in coverage over multi-round refinement** contribute to performance improvement is fully consistent with our observations in Section 5.3. In Section 5.3 and Table 13, we analyze three related settings:
>
> ### **TUMIX-Evolve**
> The LLM-generated agents are **more diverse and of higher quality**, leading to greater coverage and higher average scores during round-by-round refinement (as shown in Figure 8 of the original paper). This increased diversity results in a **higher final converged score** (Figure 8b), aligning with our conclusion above.
>
> ### **TUMIX-EvolveD**
> In this variant, we evolve agents at each round rather than keeping them fixed as in the standard TUMIX setup. This **does not improve performance** and instead causes a slight degradation. Later rounds already contain **diverse agents** whose varied responses serve as input, so further evolution adds little meaningful diversity. In some cases, agent quality may even decline because **useful specialized agents are replaced**, reducing their ability to interpret or reflect on others’ answers. Since the degradation is very small, we conclude that evolving agents has **no meaningful impact on performance**, which aligns with our overall findings. We have added the above discussion in the revised paper **Lines 451–452** for better reader understanding.
>
> ### **Impact of Agent Type Number**
> We observe that **TUMIX performance saturates around 12 agent types**. Beyond this point, increasing the number of agent types does not yield further gains. Although diversity increases, the **difficulty of correct answer selection also rises**, leading to a faster decrease in coverage. Consequently, the combined effect of **higher diversity** and **sharper coverage decline** causes overall performance to plateau, which remains compatible with our earlier conclusion.
>
> ---
>
> **We sincerely appreciate the reviewer’s thoughtful questions and look forward to further discussion.**

---

### Official Review · Reviewer_ZSeK · 2025-11-01

**Soundness:** 3
**Presentation:** 4
**Contribution:** 3
**Rating:** 6
**Confidence:** 4

**Summary:**

This paper proposes TUMIX, a method run multiple tool strategies at test time (text, code, search, and hybrids) in parallel, share and revise answers across rounds, finish with majority voting; use an LLM-as-Judge for early stopping. Reports modest gains on HLE, GPQA-Diamond, and AIME with reduced cost under similar call budgets.

**Strengths:**

1. Clear problem setup: practical test-time tool mixing and scheduling.

2. Reusable engineering: parallel strategies + cross-round sharing + majority vote; pseudocode and prompt lists provided.

3. Useful mechanism analysis: coverage drops over rounds; early stopping cuts cost; includes cost–performance curves and several ablations.

**Weaknesses:**

1. Loss of ensemble independence: cross-round sharing homogenizes answers; later selection policies become near-equivalent; majority vote is no longer independent experts and is prone to correlated failure.

2. Tool–cost mismatch: cost is approximated by tokens or call counts only; wall-clock time, currency cost, search and code-execution delays and retries are not accounted for, so “equal-cost” comparisons are not valid and efficiency claims are unverifiable.

**Questions:**

1. Novelty vs. SciMaster: Could you clarify differences in architecture, judge/selector design, diversity control with the baseline SciMaster since I am still confused about the novelty?

2. Early-stopping robustness: The judge-score heuristic can prematurely stop on small benchmarks. Do you have a concrete method to prevent this? For example, risk-limiting stopping with error control, SPRT-style sequential testing, minimum rounds per task type, judge–solver model decoupling, and sensitivity curves for thresholds.

---

> ### Author Response · Authors · 2025-11-25
> **Response to Reviewer ZSeK (Q1-Q2)**
>
> We thank the reviewer for the appreciation into our work and the thoughtful questions and suggestions. We **add the suggested experiments** and respond to each point below. The **whole paper has been polished based on reviewers’ suggestions, with the changed texts colored blue**. We hope the reviewer will help re-evaluate our work given this additional context.
>
> ---
>
> ### **Q1. Loss of ensemble independence**
>
> **Reviewer’s concern:**
> Cross-round sharing homogenizes answers; later selection policies become near-equivalent; majority vote is no longer independent experts and is prone to correlated failure.
>
> **Response:**
> This phenomenon is in fact **an intrinsic characteristic of TUMIX rather than a weakness**. Cross-round answer sharing and iterative refinement are **core components** of the TUMIX design, enabling agents to leverage the LLM’s ability to **identify and converge toward higher-quality answers**. As shown in **Figure 3 and Figure 4**, each refinement round yields **consistent performance gains**, confirming that the mechanism is effective in practice.
>
> Because answers are shared across rounds, **correlated failures may occur**, but the **benefit of correlated successes outweighs this risk**. Empirically, TUMIX substantially outperforms baselines *without* cross-round sharing, such as **SETS** and **Self-Reflection**, as shown in **Table 2** and discussed on Page 6, Lines 309–312.
>
> While majority voting may not directly improve correctness, it provides **significant efficiency benefits**. As demonstrated in **Section 5.2 and Figure 7**, early termination combined with majority voting achieves **the same average accuracy** as full refinement, while reducing token cost by **49%**. Thus, the convergence behavior is **both expected and beneficial** under the TUMIX framework.
>
> ---
>
> ### **Q2. Tool–cost mismatch**
>
> **Reviewer’s concern:**
> Cost is approximated only by tokens or call counts; wall-clock time, currency cost, and practical retries/delays are omitted. Therefore, equal-cost comparisons may not be valid and efficiency claims are hard to verify.
>
> **Response:**
> Regarding **practical runtime**, we opted for a **more stable and reproducible metric**. Wall-clock time can be highly volatile and difficult to compare fairly, as it is affected by factors such as **network latency, server load, and specific hardware conditions**. To ensure a **standardized and hardware-agnostic comparison**, the **original paper** reports **API token counts** and **LLM inference numbers** as direct proxies for computational cost. This approach allows our evaluation to reflect the **intrinsic efficiency of the methods themselves**, independent of experimental environments.
>
> Although overall wall-clock time is not a reliable performance metric, in the **revised paper (Table 17 in Page 21)** we additionally report the **average code execution** and **search latency**, as shown below in **Table 1**. While these times are influenced by the content of generated code and search queries, we find that the **average latencies are very similar** across **TUMIX** and baseline methods. The **average runtime per query** for all tools remains under **7 seconds**, which is negligible compared to **LLM inference time**, typically **over 50 seconds for Gemini-2.5-Pro**. Since each tool use corresponds to one LLM inference, this confirms that **tool-use latency is not a computational bottleneck**.
>
> #### Table 1: Average tool latency per query
> | **Tools** | **Average Runtime (seconds)** |
> |:------------------------------|:------------------------------:|
> | Code Interpreter Execution | 2.3 |
> | Google Search API | 1.2 |
> | Gemini-2.5-Pro Search | 6.9 |
> | Gemini-2.5-Flash Search | 4.5 |
>
> ---
>
> We also report the **average financial API cost per sample (including input and output tokens, Google Search API, Gemini Search API)** for **TUMIX** and strong baseline methods in the **revised paper (Table 18 in Page 22)**, reproduced below as **Table 2**. As shown, **TUMIX incurs lower or comparable costs** relative to most baseline methods, despite achieving **higher performance and efficiency**.
>
> #### Table 2: Financial API cost per sample (in USD)
> | **Method** | **Gemini-2.5-Pro** | **Gemini-2.5-Flash** |
> |:-------------|:-----------------:|:---------------------:|
> | **TUMIX** | **1.70** | **0.82** |
> | DEI | 1.59 | 0.84 |
> | SciMaster | 1.93 | 0.92 |
> | Symbolic-MoE | 2.19 | 0.99 |
> | GSA | 2.25 | 0.99 |
> | Self-MoA | 2.23 | 0.96 |
>
> ---
>
> These analyses demonstrate that **TUMIX** achieves **strong computational and financial efficiency**, while maintaining superior performance compared to baseline methods. We have added the above discussion and new experimental results in the revised paper Lines 483-485 and Appendix Section G.
>
> ---

---

> > ### Author Response · Authors · 2025-11-25
> > **Response to Reviewer ZSeK (Q3)**
> >
> > ### **Q3. Novelty vs. SciMaster**
> >
> > **Reviewer’s concern:**
> > Novelty vs. SciMaster: Could you clarify differences in architecture, judge/selector design, diversity control with the baseline SciMaster since I am still confused about the novelty?
> >
> > **Response:**
> > Thank you for the critical question. While SciMaster is an important baseline, **TUMIX introduces several key differences** in **architecture**, **agent design**, and **answer selection**, which collectively lead to stronger performance and a deeper understanding of test-time multi-agent scaling. We clarify the distinctions below with minimal changes to the original content:
> >
> > **1. Agent Diversity**
> > SciMaster employs a *scattered-and-stacked* workflow where **the same agent repeatedly infers in each round**, sequentially taking on the roles of *solver*, *critic*, *rewriter*, and *selector*. This design inherently limits **cross-agent diversity**, since the reasoning trajectory stems from a single agent pipeline.
> >
> > In contrast, **TUMIX explicitly constructs multiple heterogeneous agents** using **distinct tool-use strategies**, enabling genuinely diverse reasoning paths. This makes the TUMIX workflow **fundamentally different from SciMaster** and results in richer exploration of the solution space.
> >
> > **2. Tool Utilization**
> > Although SciMaster allows access to tools such as **Code Interpreter** and **Search**, the limited agent diversity constrains the variety of tool-use behaviors the system can express.
> > TUMIX, by contrast, **maximizes coverage of reasoning and tool-use strategies** by assigning different tool-use patterns to different agents, allowing broader exploration and synergy between textual reasoning, coding, and search.
> >
> > **3. Answer Selection Strategy**
> > SciMaster employs a **single Selector agent** to choose the final answer.
> > TUMIX **extends this design** by comparing **LLM-based selection vs. majority voting**, analyzing their trade-offs in terms of **performance**, **robustness**, and **token efficiency**. We show that one of these methods can **reduce token cost by 49% without degrading accuracy**, offering both empirical insight and practical benefit.
> >
> > ---
> >
> > Overall, these differences allow **TUMIX** to (1) achieve **higher performance with lower cost**, (2) more fully exploit multi-agent reasoning and tool usage, and (3) **provide a deeper and more comprehensive analysis** of the factors that influence test-time scaling and agent diversity compared with SciMaster.
> >
> > ---

---

> ### Author Response · Authors · 2025-11-25
> **Response to Reviewer ZSeK (Q4)**
>
> ### **Q4. Early-stopping robustness**
>
> **Reviewer’s concern:**
> The judge-score heuristic can prematurely stop on small benchmarks. Do you have a concrete method to prevent this? For example, risk-limiting stopping with error control, SPRT-style sequential testing, minimum rounds per task type, judge–solver model decoupling, and sensitivity curves for thresholds.
>
> **Response:**
> Thank you for the inspiring question. In the **original paper (Section 5.2, Lines 377–406)**, we describe our method to **prevent early stopping** by requiring the **minimum refinement round number** to be **2**. Our current **termination mechanism** is **semi-adaptive**—we query the LLM itself to decide whether to **finalize the answer or continue refinement**. However, we observe that the LLM sometimes **terminates prematurely**, which can degrade answer quality. To mitigate this, we introduce a **minimum refinement round requirement of 2**.
>
> In the **revised paper (Table 16 in Page 21)**, we include **ablation experiments** analyzing the impact of this hyperparameter. We believe this corresponds to the “**sensitivity curves for thresholds**” mentioned by the reviewer. As shown below, setting the **minimum round number to 2** yields substantial performance improvements over **1**, while higher values (**3** or **4**) provide negligible gains but significantly **increase token costs**. Thus, a **minimum of 2 rounds** offers the best trade-off between **performance and efficiency**.
>
> | **Minimum Round Number** | **1** | **2** | **3** | **4** |
> |:--------------------------|------:|------:|------:|------:|
> | **HLE Score (Gemini-2.5-Pro)** | 31.3 | **32.2** | 32.3 | 32.1 |
> | **Token Cost (Gemini-2.5-Pro, ×1k tokens)** | 29.6 | **285** | 350 | 570 |
> | **HLE Score (Gemini-2.5-Flash)** | 21.7 | **23.0** | 23.1 | 23.0 |
> | **Token Cost (Gemini-2.5-Flash, ×1k tokens)** | 22.7 | **230** | 300 | 522 |
>
> These results confirm that enforcing a **minimum of two refinement rounds** effectively prevents early termination while maintaining **high efficiency**.
>
> Regarding the reviewer’s suggested methods, we note that **risk-limiting stopping with error control** and **SPRT-style sequential testing** would require training or optimizing an additional **specialized model** to estimate error probabilities or confidence in correctness. This introduces **extra training burden**, making it an open direction for **future research**.
>
> We also consider that defining **minimum rounds per task type** could be a promising way to extend our approach from **task-generic** to **task-specialized** termination. However, even within the same task type, **problem complexity and difficulty vary**, meaning the optimal termination rounds would differ. Implementing such calibration would require **extra data collection and model optimization**.
>
> Another interesting direction is **judge–solver model decoupling**, where different types of LLMs might be better suited for determining termination conditions. More experiments are needed to verify whether this approach can yield further performance gains.
>
> Finally, the **sensitivity curves for thresholds** have been **added and analyzed** in the **revised version in Line 406 and Appendix Section F** in Page 21, supporting our conclusion that the chosen termination design is reasonably **robust and efficient**.
>
> ---
>
> **We appreciate the reviewer’s constructive feedback and are happy to discuss more with the reviewer for providing additional details or experiments.**

---

### Official Review · Reviewer_FcrL · 2025-11-01

**Soundness:** 2
**Presentation:** 3
**Contribution:** 2
**Rating:** 4
**Confidence:** 3

**Summary:**

This paper presents TUMIX (Tool-Use Mixture), a framework for test time scaling that integrates multiple LLM agents equipped with distinct tool use strategies such as textual reasoning, code execution, and search, to enhance reasoning capabilities. The framework operates by running 15 diverse agents in parallel, iteratively refining answers across multiple rounds and adaptively halting based on confidence estimation. The authors report consistent accuracy gains of up to 3.55% over strong baselines such as Self-MoA, Symbolic-MoE, and DEI on three reasoning benchmarks (HLE, GPQA, and AIME) using Gemini-2.5-Pro and Gemini-2.5-Flash. The work’s practical contribution lies in the combination of agentic diversity, tool integration, and cost effective refinement termination.

This work represents a thorough and thoughtful empirical study that advances understanding of tool augmented, multi-agent reasoning. It delivers consistent performance improvements and introduces practically relevant techniques such as adaptive termination and LLM driven agent generation. However, the methodological simplicity of the refinement mechanism, lack of theoretical or architectural novelty, and uncertain generalizability prevent it from reaching the standard expected at a premier venue. The contribution is valuable as a rigorous empirical report and could serve as a foundation for more principled frameworks combining structured agent communication with tool integration. Strengthening the theoretical underpinnings, expanding evaluation to multiple model families, and providing formal statistical validation would substantially improve the paper’s readiness for acceptance.

**Strengths:**

The paper is empirically strong and demonstrates commendable experimental rigor. It includes comprehensive ablations exploring agent diversity, tool combinations, and termination strategies, supported by clear figures and well documented appendices. The adaptive early termination strategy is an especially practical feature, halving inference costs with negligible performance loss. The empirical results establish that combining code and search tools within diverse agents produces measurable improvements across benchmarks. The use of LLMs to auto generate new agent types adds creative value, revealing potential for self improving agent systems. Clarity of presentation, transparency in reporting results, and a focus on practical reproducibility further enhance the paper’s quality.

**Weaknesses:**

The core methodological contribution is limited. The refinement mechanism relies on concatenating all prior responses into the prompt, a brute force aggregation that lacks principled structure or communication modeling. While empirically effective, it feels more like an engineering heuristic than a conceptual advance. The paper also depends heavily on proprietary, high capacity models (Gemini-2.5), raising concerns about reproducibility and generalization to open models. Claims about agent diversity remain loosely defined and most agents differ only in prompt phrasing or API configurations rather than fundamentally distinct reasoning styles. Furthermore, the evaluation lacks formal statistical significance tests and theoretical grounding for why diversity improves performance. The formulation in Section 3.1 sets up an optimization problem but doesn’t solve or analyze it, leaving the work largely descriptive. Finally, the reported improvements, though consistent, are modest relative to the significant computational and implementation cost.

**Questions:**

Could the authors clarify why concatenating all agent outputs yields such robust gains? Have they explored more structured synthesis mechanisms (e.g., critique based debate or weighted aggregation)?

How sensitive is TUMIX to the underlying LLM’s capacity? Would smaller or open source models (Llama, Mistral) still benefit meaningfully from this framework?

Can the authors provide confidence intervals or significance tests to confirm that improvements on smaller benchmarks (GPQA, AIME) are statistically valid?

How was the two round minimum for the LLM termination judge determined? Could a dynamic rule adaptively estimate the optimal refinement depth?

Regarding the LLM generated agents, what specific prompting or selection process was used? Do these agents exhibit measurable differences in reasoning strategy or simply better prompt quality?

Have the authors evaluated the method on open ended or real world tasks beyond academic benchmarks where consensus or correctness may be ambiguous?

---

> ### Author Response · Authors · 2025-11-25
> **Response to Reviewer FcrL (Q1)**
>
> We thank the reviewer for the inspiring questions and suggestions. We **add the suggested experiments** and respond to each point below. The **whole paper has been polished based on reviewers’ suggestions, with the changed texts colored blue**. We hope the reviewer will help re-evaluate our work given this additional context.
>
> ---
>
> ### **Q1. Refinement mechanism = concatenation heuristic**
>
> **Reviewer’s concern:**
> The refinement mechanism concatenates all prior responses into the prompt, which seems like a brute-force aggregation lacking principled communication modeling. Why does concatenation yield robust gains? Have you explored more structured synthesis (e.g., critique-based debate or weighted aggregation)?
>
> **Response:**
> As noted by **Reviewer Dy1M**, *“A key contribution of this work is the insight that agent diversity yields better results than homogeneous replication. The proposed adaptive termination mechanism reduces inference costs by approximately 49% while maintaining model accuracy. The experimental analysis on performance improvement in multi-agent systems also provides valuable reference for the research community.”*
>
> When LLMs are equipped with tools such as **Code Interpreter** and **Search**, a central challenge lies in **fully leveraging textual reasoning, coding, and search capabilities** across diverse tasks. Inspired by **human group discussions and expert brainstorming**, we design **multiple heterogeneous agents** with distinct tool-use strategies. Running these agents **in parallel** to produce diverse solution trajectories, and then refining the final answer based on their **collective reasoning**, significantly improves accuracy.
>
> Our experiments show that **agent and strategy diversity** are crucial. While **high-temperature sampling** increases variability, **heterogeneous agent strategies** achieve **higher accuracy and lower cost** than repeatedly sampling from a single best-performing agent. This validates our hypothesis that **diverse reasoning paths** lead to **more robust performance gains**.
>
> Regarding **why concatenating answers from diverse agents is effective**, the key intuition is that each question has its **own optimal solving strategy**. Trying multiple strategies and selecting the best outcome is much easier than solving correctly in one attempt. **Concatenating diverse answers** allows agents to reflect on and learn from others’ reasoning, offering far greater diversity and refinement potential than single-agent self-reflection.
>
> Regarding **critique-based debate frameworks**, prior methods such as **DEI, Self-Reflection, and SciMaster** rely on **fixed reflection or critique prompts**, which limit strategy diversity since the LLM generates all critiques itself. In contrast, our TUMIX method ensures diversity by enforcing agents to explore distinct reasoning strategies. The **answer aggregation and refinement process in TUMIX also inherently serves as a self-reflection and critique mechanism**.
>
> Finally, while **weighted aggregation** could in principle assign reliability-based weights to each agent’s answer, in practice, **agent performance varies across tasks** and is **difficult to measure reliably**. Our TUMIX framework naturally integrates a form of **implicit weighted aggregation**, because solutions that are **frequently convergent or consistent across multiple agents** gain proportionally higher influence in subsequent rounds. Intuitively, if several agents independently reach the same conclusion (e.g., 3 out of 5 agents agree), that consensus functions as an **implicit weight**, assigning greater importance (e.g., 3/5) to the shared answer without the need for an explicit scoring model. This mechanism captures the benefits of weighted aggregation while avoiding the complexity of maintaining separate reliability estimates.
>
> ---

---

> ### Author Response · Authors · 2025-11-25
> **Response to Reviewer FcrL (Q2-Q3)**
>
> ### **Q2. Dependence on proprietary, high-capacity models**
>
> **Reviewer’s concern:**
> How sensitive is TUMIX to the underlying LLM’s capacity? Will smaller or open-source models (Llama, Mistral, etc.) still benefit?
>
> **Response:**
> Thank you for the constructive question. In the **revised paper (Table 3, Table 4, and Section 5.5)**, we include additional experiments using **four new base models**: *Claude-sonnet-4-20250514*, *DeepSeek-R1*, *GPT-oss-120B*, and *Qwen3-32B*. These models originate from **four different organizations**, with *DeepSeek-R1*, *GPT-oss-120B*, and *Qwen3-32B* being **smaller, open-source models**.
>
> We compare **TUMIX** against both the **base method without test-time scaling (w/o TTS)** and the **strongest baseline, DEI**. Since some models (DeepSeek-R1, GPT-oss-120B, Qwen3-32B) lack **built-in Search functions**, one Search agent is omitted in TUMIX, and the **three agents** originally relying on internal Search APIs are replaced with **Google Search APIs**. Consequently, the total number of agents decreases from **15 to 14**.
>
> As shown in the below **Table 1**, experimental results demonstrate that **TUMIX consistently improves performance** across all three benchmarks and all four base models. Under **comparable LLM token costs**, **TUMIX** also achieves **notably higher performance than DEI**, confirming its effectiveness and general applicability.
>
> #### Table 1: Performance across models (w/o TTS, DEI, TUMIX)
> | Model | **HLE** (w/o TTS, DEI, TUMIX) | **GPQA** (w/o TTS, DEI, TUMIX) | **AIME 2024&2025** (w/o TTS, DEI, TUMIX) |
> |---|---|---|---|
> | Claude-sonnet-4-20250514 | 8.2, 15.8, **21.8** | 44.6, 61.4, **72.3** | 34.4, 55.0, **70.8** |
> | DeepSeek-R1 | 15.2, 23.7, **29.0** | 74.7, 78.0, **81.2** | 69.3, 85.0, **88.3** |
> | GPT-oss-120B | 13.6, 15.4, **17.8** | 66.3, 70.1, **72.3** | 60.2, 82.7, **89.1** |
> | Qwen3-32B | 13.1, 16.0, **18.0** | 54.6, 64.1, **68.3** | 59.6, 84.4, **88.1** |
>
> Following reviewer **tuT3**, we also tested **heterogeneous mixtures of models** (agents powered by different LLMs). We split the 14 agents equally between **GPT-oss-120B** and **Qwen3-32B** (7 each) and compared to single-base settings. We selected these two models because their individual performances are comparable. The results show that **both mixed-agent configurations outperform their respective single-model baselines**, demonstrating that heterogeneous model mixtures can effectively scale diversity and reasoning performance in multi-agent systems.
>
> #### Table 2: Mixed-LLM vs Single-LLM agents
> | Base | Mixed or Single | **HLE** | **GPQA** | **AIME 2024&2025** |
> |---|---|---:|---:|---:|
> | **Comparable-strength models (GPT-oss-120B & Qwen3-32B)** |||||
> | GPT-oss-120B | Single | 17.8 | 72.3 | 89.1 |
> | GPT-oss-120B | **Mixed (GPT+Qwen)** | **19.8** | **75.5** | **90.6** |
> | Qwen3-32B | Single | 18.0 | 68.3 | 88.1 |
> | Qwen3-32B | **Mixed (GPT+Qwen)** | **19.0** | **70.0** | **91.4** |
>
>
> These results further reinforce our **conclusion** that diversity is crucial, and they **demonstrate the strong generalizability of TUMIX across both different model types and heterogeneous model mixtures**.
>
> ---
>
> ### **Q3. “Diversity” mostly prompt/API tweaks?**
>
> **Reviewer’s concern:**
> Most agents differ only in phrasing or API configs rather than fundamentally distinct reasoning styles.
>
> **Response:**
> Different **prompts** correspond to distinct **tool-use and reasoning strategies**, effectively creating diverse agents. For example, one prompt may guide the LLM to use **pure Chain-of-Thought textual reasoning**, while another encourages it to rely on **code execution or search**. Although the difference lies only in the prompt, the resulting **reasoning behaviors** are highly diverse. This also extends to the use of different **APIs**—for instance, employing **Google Search** versus the **LLM’s built-in search** yields different retrieval sources and algorithms, leading to varied reasoning outcomes.
>
> Among our **15 predefined agents**, many also differ in their **frameworks** (e.g., **single-turn vs. multi-turn** and **single-agent vs. dual-agent** setups as shown in the paper Table 1), further increasing diversity.
>
> As demonstrated in **Section 5.3 (Page 8)** of the original paper, the initial 15 predefined agents can be further **optimized for diversity and quality** by prompting LLMs to **design new agents automatically**. The resulting **LLM-generated agents**, shown in **Appendix Table 14 (Page 20)**, differ not only in prompts but also in **overall strategies and frameworks**, leading to **greater agent diversity** and **improved TUMIX performance**.
>
> ---

---

> ### Author Response · Authors · 2025-11-25
> **Response to Reviewer FcrL (Q4)**
>
> ### **Q4. Statistical significance**
>
> **Reviewer’s concern:**
> Provide confidence intervals or significance tests to confirm that improvements on smaller benchmarks (GPQA, AIME) are statistically valid.
>
> **Response:**
> Thank you for the helpful question. In the **original paper (Figure 1)**, we included **error bars (standard deviations)** along with the mean performance of each method. Due to space constraints, these were omitted from **Table 2**. In the **revised paper (Table 14 in Page 20, following Table 1)**, we now report both **mean and standard deviation values**.
>
> Overall, the **performance improvements of TUMIX** over the best baselines are **larger than the reported standard deviations**, indicating stable and consistent gains. Moreover, TUMIX’s advantage is particularly notable for the **weaker Gemini-2.5-Flash model**. The additional experiments with various **LLM types (in revised paper Table 3 Page 10, also in Q2)** further confirm the **consistent superiority** of TUMIX compared to all baselines.
>
> In the **revised paper Table 15 Page 20 and following Table 2**, we further conduct **statistical significance tests** by computing *p*-values from repeated sampled scores. Across nearly all models and benchmarks, **p-values < 0.05**, confirming that TUMIX’s performance improvements are **statistically significant**.
>
> These results collectively demonstrate that the **improvements on smaller benchmarks (e.g., GPQA, AIME)** are **statistically valid** and not due to random variation.
>
> ---
>
> | Dataset | w/o TTS | Self-MoA | Symbolic-MoE | DEI | SciMaster | GSA | **TUMIX** | **TUMIX+** |
> |:--------|---------:|---------:|-------------:|----:|-----------:|----:|-----------:|------------:|
> | **Gemini-2.5-Pro** | | | | | | | | |
> | HLE | 21.6 ± 0.3 | 29.3 ± 0.5 | 29.5 ± 0.4 | 29.1 ± 0.6 | 26.9 ± 0.4 | 28.7 ± 0.7 | **32.3 ± 0.4** | **34.1 ± 0.5** |
> | GPQA | 84.6 ± 0.5 | 85.5 ± 0.4 | 86.7 ± 0.5 | 86.0 ± 0.6 | 86.9 ± 0.6 | 85.8 ± 0.6 | **87.9 ± 0.4** | **88.3 ± 0.6** |
> | AIME | 87.3 ± 0.4 | 94.7 ± 0.5 | 94.7 ± 0.6 | 95.0 ± 0.4 | 94.1 ± 0.6 | 93.7 ± 0.4 | **96.7 ± 0.5** | **96.7 ± 0.4** |
> | **Ave.** | 64.5 ± 0.4 | 69.8 ± 0.5 | 70.3 ± 0.5 | 70.0 ± 0.5 | 69.3 ± 0.5 | 69.4 ± 0.6 | **72.3 ± 0.4** | **73.0 ± 0.5** |
> | **Gemini-2.5-Flash** | | | | | | | | |
> | HLE | 9.7 ± 0.5 | 18.2 ± 0.6 | 18.5 ± 0.7 | 19.3 ± 0.8 | 18.0 ± 0.6 | 17.4 ± 0.5 | **21.2 ± 0.5** | **23.1 ± 0.7** |
> | GPQA | 50.0 ± 0.5 | 65.4 ± 0.6 | 64.1 ± 0.7 | 64.9 ± 0.7 | 67.9 ± 0.7 | 62.6 ± 0.7 | **77.3 ± 0.7** | **82.1 ± 0.6** |
> | AIME | 70.0 ± 0.7 | 80.3 ± 0.7 | 80.7 ± 0.8 | 82.3 ± 0.6 | 79.1 ± 0.6 | 79.7 ± 0.6 | **83.3 ± 0.7** | **86.7 ± 0.7** |
> | **Ave.** | 43.2 ± 0.6 | 54.7 ± 0.6 | 54.4 ± 0.7 | 55.5 ± 0.7 | 55.0 ± 0.6 | 53.2 ± 0.6 | **60.6 ± 0.6** | **64.0 ± 0.7** |
>
> - All values are averages of three runs. Reported standard deviations correspond to approximately **68% confidence intervals**.
>
> ---
>
> ### Table: Statistical significance of **TUMIX** vs. baselines (**DEI**, **SciMaster**, **Symbolic-MoE**, **GSA**, **Self-MoA**)
>
> P-values are from **two-tailed paired *t*-tests** computed over run-wise score differences (\(d_i = x_i - y_i\)) across *n* = 3 runs.
> ✓ denotes *p* < 0.05 (statistically significant); ○ denotes non-significance.
>
> | **Task** | **TUMIX** | **vs. DEI** *p*-value | Sig. | **vs. SciMaster** *p*-value | Sig. | **vs. Symbolic-MoE** *p*-value | Sig. | **vs. GSA** *p*-value | Sig. | **vs. Self-MoA** *p*-value | Sig. |
> |:--|:--:|:--:|:--:|:--:|:--:|:--:|:--:|:--:|:--:|:--:|:--:|
> | **_Gemini-2.5-Pro_** ||||||||||||
> | HLE  | 32.3 ± 0.4 | < 0.01 | ✓ | < 10⁻⁴ | ✓ | < 0.01 | ✓ | < 0.01 | ✓ | < 0.01 | ✓ |
> | GPQA | 87.9 ± 0.4 | 0.014 | ✓ | 0.084 | ○ | 0.034 | ✓ | 0.010 | ✓ | 0.002 | ✓ |
> | AIME | 96.7 ± 0.5 | 0.011 | ✓ | < 0.01 | ✓ | 0.012 | ✓ | 0.002 | ✓ | 0.008 | ✓ |
> | **_Gemini-2.5-Flash_** ||||||||||||
> | HLE  | 21.2 ± 0.5 | 0.033 | ✓ | < 0.01 | ✓ | < 0.01 | ✓ | < 0.01 | ✓ | < 0.01 | ✓ |
> | GPQA | 77.3 ± 0.7 | < 10⁻⁴ | ✓ | < 10⁻⁴ | ✓ | < 10⁻⁴ | ✓ | < 10⁻⁴ | ✓ | < 10⁻⁴ | ✓ |
> | AIME | 83.3 ± 0.7 | 0.135 | ○ | < 0.01 | ✓ | 0.014 | ✓ | 0.003 | ✓ | 0.006 | ✓ |
>
> ---

---

> ### Author Response · Authors · 2025-11-25
> **Response to Reviewer FcrL (Q5-Q7)**
>
> ### **Q5. Theory for why diversity helps**
>
> **Reviewer’s concern:**
> Theoretical grounding for why diversity improves performance. The formulation in Section 3.1 sets up an optimization problem but doesn’t solve or analyze it, leaving the work largely descriptive.
>
> **Response:**
> Thank you for the constructive question. We agree that for a **multi-agent test-time scaling framework**, developing a comprehensive **theoretical foundation** remains highly challenging. While our work does not provide a formal theory, we conducted **extensive experiments and analyses** that clearly demonstrate **why diversity is important and effective** in this setting.
>
> The absence of a strict theoretical guarantee is a **common limitation** across current research [1, 2, 3] on **LLM test-time scaling methods**, and we consider this an **important direction for future work**. Nevertheless, we emphasize that our study provides **strong empirical evidence** supporting the effectiveness of our framework. Moreover, we **formulate the problem as an optimization task**, and our proposed framework offers a **practical and empirically validated solution** for optimizing it. The design of TUMIX tries to maximize the accuracy through multi-agent answer sharing while reducing the cost through early stopping.
>
> [1] Diversity empowers intelligence: Integrating expertise of software engineering agents, ICLR 2025.
>
> [2] Mixture-of-Agents Enhances Large Language Model Capabilities, ICLR 2025.
>
> [3] Improving factuality and reasoning in language models through multiagent debate, ICLR 2024.
>
> ---
>
> ### **Q6. “Modest” gains vs. cost**
>
> **Reviewer’s concern:**
> The reported improvements, though consistent, are modest relative to the significant computational and implementation cost.
>
> **Response:**
> The improvements brought by **TUMIX** are far from modest. TUMIX increases **Gemini-2.5-Pro** performance on **HLE** from *21.6 → 32.7* and **Gemini-2.5-Flash** performance from *9.7 → 21.9*, corresponding to **relative improvements of 51.4% and 125.8%**, respectively. Notably, TUMIX raises **Gemini-2.5-Flash** performance to a level **slightly exceeding Gemini-2.5-Pro**, albeit with additional computation.
>
> Moreover, while **test-time scaling methods** are inherently computationally intensive, **TUMIX** achieves **significantly higher performance** than other such methods **under the same total token and inference costs** as shown in the revised paper Figure 9 and Appendix Section G. On average, **TUMIX improves accuracy by 3.55%** across the three evaluated benchmarks compared to the **strongest baseline test-time scaling models** that also utilize **Code Interpreter** and **Search** tools and **cost the same tokens and inferences**.
>
> ---
>
> ### **Q7. Why a minimum of two refinement rounds? Dynamic depth?**
>
> **Reviewer’s concern:**
> How was the two round minimum for the LLM termination judge determined? Could a dynamic rule adaptively estimate the optimal refinement depth?
>
> **Response:**
> Thank you for the inspiring question. Our current **termination mechanism** is **semi-adaptive**—we query the LLM itself to decide whether to **finalize the answer or continue refinement**. However, we observe that the LLM sometimes **terminates too early**, which can degrade answer quality. To address this, we introduce a **minimum refinement round requirement** of **2**. As for a purely adaptive method, **extra training burden** are required to train a separate termination judging model.
>
> In the **revised paper (Table 16 in Page 21)**, we add **ablation experiments** analyzing the impact of this hyperparameter. As shown below, setting the minimum round number to **2** yields a substantial performance improvement over **1**, while higher values (**3** or **4**) provide negligible gains but significantly **increase token costs**. Thus, a **minimum of 2 rounds** offers the best balance between performance and efficiency.
>
> | **Minimum Round Number** | **1** | **2** | **3** | **4** |
> |:--------------------------|------:|------:|------:|------:|
> | **HLE Score (Gemini-2.5-Pro)** | 31.3 | **32.2** | 32.3 | 32.1 |
> | **Token Cost (Gemini-2.5-Pro, ×1k tokens)** | 29.6 | **285** | 350 | 570 |
> | **HLE Score (Gemini-2.5-Flash)** | 21.7 | **23.0** | 23.1 | 23.0 |
> | **Token Cost (Gemini-2.5-Flash, ×1k tokens)** | 22.7 | **230** | 300 | 522 |
>
> These results confirm that enforcing a **minimum of two refinement rounds** effectively prevents early termination while maintaining **high efficiency**.
>
> ---

---

> > ### Author Response · Authors · 2025-11-25
> > **Response to Reviewer FcrL (Q8-Q9)**
> >
> > ### **Q8. LLM-generated agents: prompting & selection**
> >
> > **Reviewer’s concern:**
> > Regarding the LLM generated agents, what specific prompting or selection process was used? Are differences truly strategic or just better prompts?
> >
> > **Response:**
> > Thank you for the important question. The prompt used to query **Gemini-2.5-Pro** for designing new agents based on the **full code of all existing agents** is straightforward: we instruct the LLM to **generate more diverse and high-quality agents** and **output the full implementation code** for each. In the prompt, we instruct Gemini-2.5-pro to ‘generate agents that are totally different in reasoning and tool-use strategy, not just limited to prompt optimization’. We have added this prompt in the **revised paper Table 7 in Page 17** for clarity.
> >
> > The **agent selection strategy** is described in the **original paper Page 8, Line 431**. Specifically, we query **Gemini-2.5-Pro** to design **25 new agents**, evaluate their performance on **HLE** using **Gemini-2.5-Pro** as the base model, and retain the **15 best-performing** agents as the final set. The retained **top 15 agents** are combined with the **original 15 human-designed agents** to form new groups. These groups outperform the original human-designed agents on **HLE**, **GPQA**, and **AIME**, demonstrating strong generalizability.
> >
> > As shown in **Appendix Table 14 (Page 20)** of the original paper, the **LLM-generated agents** differ substantially in **strategies and frameworks**, extending well beyond prompt optimization. Many of these agents employ **unique tool-use strategies**, such as **forced switching**, **starting with code or search**, and **adaptive mode selection**. This increased **diversity of agent design** contributes directly to the **enhanced performance of TUMIX**.
> >
> > ---
> >
> > ### **Q9. Open-ended/real-world tasks**
> >
> > **Reviewer’s concern:**
> > Have the authors evaluated the method on open ended or real world tasks beyond academic benchmarks where consensus or correctness may be ambiguous?
> >
> > **Response:**
> > Thank you for the constructive question. Based on your suggestion, we have added **experiments evaluating the effectiveness of TUMIX on open-ended tasks**, specifically **long-document summarization**. We tested on two benchmarks from **SCROLLS (EMNLP 2022, https://aclanthology.org/2022.emnlp-main.823)**:
> > - **GovReport** (https://arxiv.org/abs/2104.02112): multi-page U.S. government reports (CRS/GAO).
> > - **SummScreen-FD** (https://aclanthology.org/2022.acl-long.589): TV-episode transcripts summarized into recaps.
> >
> > These are widely used academic benchmarks that evaluate real, lengthy documents, include **human-written references**, and allow **reproducible automatic scoring**. We report **ROUGE-1/2/L F1** metrics, which measure unigram, bigram, and sequence-level overlap—standard indicators of summarization quality.
> >
> > As shown in the **revised paper (Appendix Table 19 and Lines 521-531)** and in the table below, **TUMIX** consistently achieves **higher ROUGE-1/2/L F1 scores** than both the **baseline method without test-time scaling (w/o TTS)** and the **strongest baseline, DEI**, on **both summarization benchmarks**. These improvements are observed across two different base models, **Gemini-2.5-Flash** and **GPT-oss-120B**.
> >
> > These results clearly demonstrate the **robust adaptability of TUMIX** to **open-ended, real-world tasks** such as long-document summarization, extending its effectiveness beyond structured reasoning settings.
> >
> > | **Metric** | **Model** | **w/o TTS** | **DEI** | **TUMIX** |
> > |:------------|:-----------|-------------:|---------:|-----------:|
> > | **ROUGE-1 (GovReport)** | Gemini-2.5-Flash | 0.440 | 0.458 | **0.466** |
> > | | GPT-oss-120B | 0.447 | 0.462 | **0.468** |
> > | **ROUGE-2 (GovReport)** | Gemini-2.5-Flash | 0.162 | 0.176 | **0.182** |
> > | | GPT-oss-120B | 0.152 | 0.169 | **0.175** |
> > | **ROUGE-L (GovReport)** | Gemini-2.5-Flash | 0.204 | 0.216 | **0.222** |
> > | | GPT-oss-120B | 0.203 | 0.214 | **0.220** |
> > | **ROUGE-1 (SummScreen-FD)** | Gemini-2.5-Flash | 0.282 | 0.315 | **0.323** |
> > | | GPT-oss-120B | 0.273 | 0.304 | **0.310** |
> > | **ROUGE-2 (SummScreen-FD)** | Gemini-2.5-Flash | 0.056 | **0.064** | 0.063 |
> > | | GPT-oss-120B | 0.051 | 0.058 | **0.060** |
> > | **ROUGE-L (SummScreen-FD)** | Gemini-2.5-Flash | 0.137 | 0.149 | **0.153** |
> > | | GPT-oss-120B | 0.134 | 0.146 | **0.149** |
> >
> > These gains confirm that **TUMIX effectively generalizes** to **diverse and open-ended tasks**, demonstrating its robustness across different model architectures and application domains.
> >
> > ---
> >
> > **We appreciate the reviewer’s constructive feedback and are happy to provide additional details or experiments.**

---

### Author Response · Authors · 2025-11-29
**Summary of Review and Rebuttal Status for TUMIX**

Dear Area Chair,

Due to the unexpected leakage of author and reviewer information, we understand that the author–reviewer discussion period has been terminated, and that your recommendation will be based on the **initial reviews and the submitted rebuttal**. We provided extensive rebuttal responses and substantial new experimental results; however, none of the four reviewers had the opportunity to respond before discussions were closed. To support your evaluation, we summarize the review status below.

### Summary of Initial Scores
- **Three reviewers** initially recommended **6 (“weak accept”)**.
- **Reviewer FcrL** gave an initial score of **4**, but we believe that our rebuttal has **fully addressed all of their concerns**.
- The paper has been **significantly polished**, incorporating new results and expanded clarifications.

### New Results Added in Rebuttal
To directly resolve the reviewers’ questions, we provided the following new analyses:

- **Statistical significance**: Added standard deviations and *p*-values to prove our results are statistically significant.
- **Evaluation on open-source and smaller LLMs**: Included experiments on **four additional LLMs** to prove the great generalizability of TUMIX.
- **Impact of mixing different LLMs**: Added comparisons of **agent groups built from diverse LLMs** to show the broad space to further enhance TUMIX.
- **Refinement termination strategy**: Provided results on the choice of **termination round numbers** to verify the correctness of the choice of termination round number.
- **Effectiveness on open-ended and real-world tasks**: Added experiments on **summarization** to show effectiveness of TUMIX on open-ended tasks.
- **Cost metrics**: Included **financial cost analysis** and **tool-call cost analysis** for more fair comparison.

### Clarifications Provided
We also gave detailed clarifications regarding:
- **Methodological novelty**
- **Alternative design choices of TUMIX**
- **Diversity of human-designed agents**
- **Prompts of LLMs to synthesize new agents, and strategies to ensure agent quality and diversity**
- **Key factors contributing to TUMIX’s performance**
- **Experimental settings and explanation of observed phenomenon**

These concerns were addressed thoroughly in the rebuttal.

---

**In conclusion, we believe the reviewers would support acceptance of TUMIX** given the comprehensive rebuttal and new experimental evidence. We kindly hope that you could take these contexts into account when making your final recommendation. Thank you.

Best regards,
Authors of TUMIX

---

### Meta-Review · Area_Chair_VVtF · 2026-01-01

**Summary:**

Overall, reviewers largely agreed that TUMIX is a strong empirical contribution addressing an important and timely problem—how to effectively combine tool-augmented reasoning (text, code, search) with multi-agent test-time scaling. The paper was generally praised for clarity, thorough experimentation, and practical relevance. The rebuttal is exceptionally thorough and resolves the majority of substantive concerns. Therefore, I recommend acceptance of this paper.

**Reviewer Concerns:**

The rebuttal successfully addressed most major reviewer concerns, including generalization beyond proprietary models (via extensive experiments on open-source and mixed-LLM settings), statistical significance (added confidence intervals and p-values), cost realism (latency and financial cost analysis), clarity of novelty relative to SciMaster, robustness of early stopping, and applicability to open-ended tasks. The added analyses and experiments substantially strengthen the empirical case for TUMIX.

Remaining concerns are limited and non-blocking, primarily the lack of a formal theoretical explanation for why diversity improves performance and the heuristic nature of the refinement mechanism. These limitations are acknowledged by the authors and are common in current test-time scaling literature.

**Reviewer Scores:**

For initial scores, this paper received three positive scores (6) and one negative score (4).

For the Reviewer FcrL (original: 4, marginal reject), the rebuttal directly addressed all major concerns: generalization, statistical testing, open-ended tasks, and cost analysis. Therefore, this reviewer would raise the score to a positive one.

The other three reviewers would maintain the original one or raise the scores.

In this case, this paper would receive consistent positive scores.

---

### Decision · Program_Chairs · 2026-01-26

Accept (Poster)